# A sensory cell diversifies its output by varying Ca$^{2+}$ influx-release coupling among active zones

Özge D Özçete[1,2,3,4] (iD) & Tobias Moser[1,2,3,4,5,6,*] (iD)

## Abstract

The cochlea encodes sound pressures varying over six orders of magnitude by collective operation of functionally diverse spiral ganglion neurons (SGNs). The mechanisms enabling this functional diversity remain elusive. Here, we asked whether the sound intensity information, contained in the receptor potential of the presynaptic inner hair cell (IHC), is fractionated via heterogeneous synapses. We studied the transfer function of individual IHC synapses by combining patch-clamp recordings with dual-color Rhod-FF and iGluSnFR imaging of presynaptic Ca$^{2+}$ signals and glutamate release. Synapses differed in the voltage dependence of release: Those residing at the IHC' pillar side activated at more hyperpolarized potentials and typically showed tight control of release by few Ca$^{2+}$ channels. We conclude that heterogeneity of voltage dependence and release site coupling of Ca$^{2+}$ channels among the synapses varies synaptic transfer within individual IHCs and, thereby, likely contributes to the functional diversity of SGNs. The mechanism reported here might serve sensory cells and neurons more generally to diversify signaling even in close-by synapses.

**Keywords** calcium channel; exocytosis; nanodomain; synaptic heterogeneity; wide dynamic range coding
**Subject Category** Neuroscience
**The EMBO Journal (2021) 40: e106010**
See also: **ME Gómez-Casati & JD Goutman** (March 2021)

## Introduction

Neural systems employ functional diversity to achieve the complexity of behavior. Diversity is implemented at several levels, i.e., circuits, neurons, and subcellular functional units such as synapses. Sensory systems employ such multiscale diversity as well as adaptation to deal with the challenge of encoding a wide range of stimulus intensities (Kandel *et al*, 2012). The auditory system copes with processing a wide range of sound intensities by employing two types of sensory cells: outer hair cells to actively amplify and compress the range of mechanical inputs (Ashmore, 2008) and IHCs to adaptively encode at synapses with primary auditory neurons (type I spiral ganglion neurons [SGNs]) (Moser *et al*, 2019). Based on their physiology, these neurons can be classified into three functional subtypes, namely low, medium, and high spontaneous rate (SR) SGNs differing in the threshold and dynamic range of sound encoding (Kiang *et al*, 1965; Sachs & Abbas, 1974; Liberman, 1978; Winter *et al*, 1990; Taberner & Liberman, 2005). This functional SGN diversity appears at all tonotopic places of the cochlea, and the subtypes can even innervate the same IHC (Liberman, 1982).

Functional diversity of SGNs relates to the heterogeneity of their molecular profile, morphology, afferent, and efferent synaptic connectivity. In the cat (Liberman, 1982), back-tracing experiments linked morphology to function and showed that low-SR SGNs have thinner radial fibers (peripheral neurites) with fewer mitochondria than the high-SR ones. Low-SR SGNs preferentially innervate the modiolar (or neural) side of the IHC, where they face larger and more complex presynaptic active zones (AZs) (Liberman, 1980, 1982; Merchan-Perez & Liberman, 1996; Kantardzhieva *et al*, 2013). Larger and more complex AZs at the modiolar side of IHCs were also found in mouse (Liberman *et al*, 2011; Ohn *et al*, 2016; Michanski *et al*, 2019), guinea pig (Furman *et al*, 2013; Song *et al*, 2016), and gerbil (Zhang *et al*, 2018). In the mouse, RNA sequencing of individual SGNs indicated three distinct molecular profiles (Sun *et al*, 2018; Shrestha *et al*, 2018; Petitpré *et al*, 2018) that were suggested to correspond to low, medium, and high-SR SGNs based on the spatial segregation of their IHC innervation.

In mouse IHCs, AZ size correlates with the number of Ca$^{2+}$ channels (approximately 30 to 300) (Neef *et al*, 2018) and consequently with the maximal Ca$^{2+}$ influx at the AZ (Frank *et al*, 2009; Ohn *et al*, 2016; Neef *et al*, 2018). Should the analogy to the innervation pattern in the cat cochlea apply, it is odd that Ca$^{2+}$-triggered glutamate release from modiolar AZs, with large size and Ca$^{2+}$ influx, was to drive low-SR SGNs. A possible solution to this conundrum came from the finding that Ca$^{2+}$ influx at the modiolar AZs requires stronger depolarization than at the pillar ones (Ohn *et al*, 2016). In other words, Ca$^{2+}$ channels at modiolar AZs would be mostly closed at the IHC resting potential and require stronger receptor potentials

1   Institute for Auditory Neuroscience and InnerEarLab, University Medical Center Göttingen, Göttingen, Germany
2   Collaborative Research Center 889, University of Göttingen, Göttingen, Germany
3   Auditory Neuroscience Group, Max Planck Institute of Experimental Medicine, Göttingen, Germany
4   Göttingen Graduate Center for Neurosciences, Biophysics and Molecular Biosciences, University of Göttingen, Göttingen, Germany
5   Synaptic Nanophysiology Group, Max Planck Institute of Biophysical Chemistry, Göttingen, Germany
6   Multiscale Bioimaging Cluster of Excellence (MBExC), University of Göttingen, Göttingen, Germany
    *Corresponding author. Tel: +49 551 39 63070; E-mail: tmoser@gwdg.de

to activate, which could explain the low SR and high thresholds of their postsynaptic SGNs.

Whether and how such heterogeneous properties of presynaptic $Ca^{2+}$ signaling relate to glutamate release and to SGN firing remains to be elucidated. Exocytosis of readily releasable synaptic vesicles (SVs) in mature mouse IHCs relates near-linearly to $Ca^{2+}$ influx when varying the number of open $Ca^{2+}$ channels (Brandt *et al*, 2005; Wong *et al*, 2014; Pangrsic *et al*, 2015). Similar findings were found in mouse vestibular hair cells (Dulon *et al*, 2009). Hence, one would assume that the heterogeneous $Ca^{2+}$ signaling propagates into a concomitant diversity of transmitter release. However, this remains to be studied at the single-synapse level ideally for several AZs of a given IHC. In fact, a recent study of cerebellar synapses highlighted how differences in $Ca^{2+}$ channel–release coupling diversify synaptic transfer (Rebola *et al*, 2019).

Here, we studied the synaptic transfer function and underlying $Ca^{2+}$ dependence of release at individual IHC-SGN synapses by combining IHC patch-clamp with imaging of synaptic $Ca^{2+}$ influx and glutamate release. To detect glutamate release, we utilized the fluorescent glutamate reporter iGluSnFR (Marvin *et al*, 2013) that we targeted to the postsynaptic SGNs. Our results suggest that IHCs vary the voltage dependence of $Ca^{2+}$ channels as well as their control of release sites among their AZs. This likely enables IHCs to signal the information contained in the receptor potential into complementary neural channels for encoding the entire audible range of sound intensities.

## Results

### Optical detection of glutamate release at individual inner hair cell synapses

Fluorescence imaging allows analysis of individual IHC AZs (Griesinger *et al*, 2005; Frank *et al*, 2009; Ohn *et al*, 2016) due to their large nearest neighbor distance (~ 2 μm) (Meyer *et al*, 2009). To image glutamate release, we targeted iGluSnFR (Marvin *et al*, 2013) to the postsynaptic SGN membrane. We injected the round window of WT mice at postnatal days (P)5–7 with adeno-associated virus (AAV9, human synapsin promoter) to drive largely uniform SGN expression of iGluSnFR with several transduced afferent boutons per IHC (Fig 1A, Appendix Fig S1).

Using apicocochlear organs of Corti, acutely dissected after the onset of hearing (P15–19), we patch-clamped IHCs and simultaneously imaged postsynaptic iGluSnFR fluorescence by spinning disk confocal microscopy (Ohn *et al*, 2016). Figure 1B shows two exemplary IHCs innervated by iGluSnFR-expressing afferent boutons at their modiolar (IHC1) or pillar (IHC2) side in the given confocal sections. Sizable changes in iGluSnFR fluorescence ($\Delta$F-iGluSnFR) were evoked by brief (10-ms-long) step depolarizations to −22 mV (Fig 1B, see Appendix Fig S2 for the iGluSnFR-region of interest [ROI] detection routine). Toward the end of the perforated patch recording, we ruptured the membrane patch and introduced a TAMRA-conjugated dimeric ribbon-binding peptide to identify individual AZs (Fig 1B). When imaging, we purposely avoided the basal cap of the IHCs in which separating individual postsynaptic boutons is more challenging given the high synapse density (Meyer *et al*, 2009; Liberman *et al*, 2011).

Next, we probed the effect of the imaging plane on the iGluSnFR signal. We applied 50-ms-long step depolarizations in seven different planes each separated from the next one by 0.5 μm (ruptured patch-clamp, 10 mM intracellular EGTA, 5 mM $[Ca^{2+}]_e$, $n = 10$ boutons, $N = 6$ IHCs from four mice; Appendix Fig S3). We compared $\Delta$F-iGluSnFR in the optimal plane (the one with highest signal) to the ones in planes ± 0.5 μm and ± 1 μm from the optimal plane. We found that the $\Delta$F-iGluSnFR was rather robust toward missing the optimal plane: There was a 24.79 ± 4.33 % reduction in the peak of the iGluSnFR signal for the "± 0.5 μm planes" and a 26.83 ± 4.05 % reduction for the "± 1 μm planes". Furthermore, we checked the variability of the $\Delta$F-iGluSnFR at a given synapse. We probed the peak of the iGluSnFR signal by repetitive 20-ms-long step depolarizations from the holding potential of −87 to −17 mV applied every 20 seconds over 5 min (Appendix Fig S4, ruptured patch-clamp, 10 mM intracellular EGTA, 5 mM $[Ca^{2+}]_e$, $n = 5$ boutons, $N = 2$ IHCs from two mice). We observed a mild rundown of the $\Delta$F-iGluSnFR over repetitive stimulation: The mean peak $\Delta$F amplitude of the last three points was 36.75 ± 7.03 % smaller than the mean of the first three points (14 step depolarizations over 5 min).

To probe the specificity of $\Delta$F-iGluSnFR for reporting $Ca^{2+}$-mediated glutamate release, we tested the effect of the $Ca^{2+}$ channel blocker $Zn^{2+}$. $\Delta$F-iGluSnFR triggered by step depolarizations gradually decreased and partially recovered upon $Zn^{2+}$ application and wash-out, respectively (Fig EV1A and B). To assess potential adverse effects of iGluSnFR expression on auditory signaling, we recorded auditory brainstem responses (ABR) at P29 (~ 23 days after the AAV injection). ABR waveforms and thresholds of the injected and non-injected (control) ears were comparable (Fig EV1C) despite the efficient transduction and iGluSnFR expression of SGNs. In conclusion, AAV-mediated expression of iGluSnFR in SGNs is suitable for studying IHC glutamate release with high specificity and does not obviously alter auditory physiology.

### Deciphering synaptic glutamate release from IHC active zones

To compare IHC exocytosis on a single-synapse versus whole-cell level, we measured synaptic $\Delta$F-iGluSnFR simultaneously with well-established whole-cell membrane capacitance changes ($\Delta C_m$) (Moser & Beutner, 2000). To assess the sensitivity and saturation of $\Delta$F-iGluSnFR, we applied stimuli of different durations (2–100 ms, in pseudo-randomized order) in near-physiological conditions (perforated patch configuration, 1.3 mM extracellular $Ca^{2+}$ concentration ($[Ca^{2+}]_e$); Fig EV1D). $\Delta$F-iGluSnFR became significant at 2 ms ($P < 0.0001$), while $\Delta C_m$ were detectable only at 5 ms ($P = 0.004$; Appendix Fig S5). To evaluate a potential saturation of iGluSnFR by glutamate release, we related both the peak and the area under the curve of the iGluSnFR signal (hereafter referred to as iGluSnFR-AUC) to the corresponding $\Delta C_m$. Both measures showed a positive correlation with $\Delta C_m$ (Pearson's $r = 0.63$, $P < 0.0001$ and $r = 0.66$, $P < 0.0001$; Fig EV1F and G), indicating their robust report of exocytosis for depolarizations up to at least 100 ms. Furthermore, the decay time constant of iGluSnFR signal increased with depolarization duration (Fig EV1I). Different from postsynaptic $\Delta$F-iGluSnFR, $\Delta C_m$ also reports extrasynaptic exocytosis (Pangrsic *et al*, 2015), likely contributing to the sublinear relationship of both measures for longer stimuli.

As a complementary approach to validate ΔF-iGluSnFR as a read-out of exocytosis, we probed it by applying brief (10-ms) step depolarizations from the holding potential (−87 mV) to −57 mV in 10 mV increments up to 23 mV (applied in pseudo-randomized order; Appendix Fig S6) and simultaneously recorded $\Delta C_m$. Both the peak and the AUC of iGluSnFR signal positively correlated with the $\Delta C_m$ in the negative voltage range (from −57 to −17 mV; Pearson's $r = 0.57$, $P < 0.0001$, Pearson's $r = 0.59$, $P < 0.0001$, respectively)

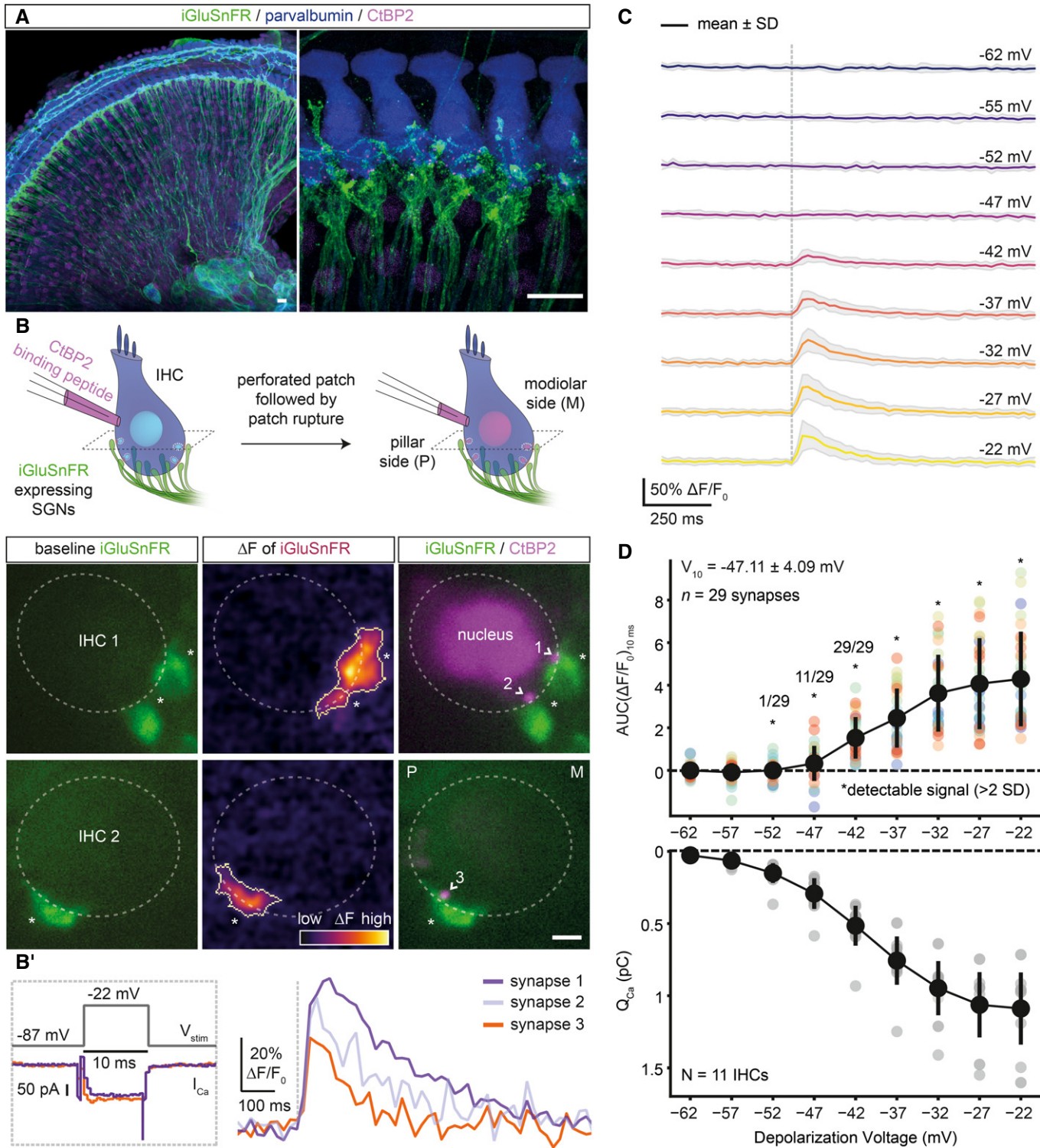

**Figure 1.**

**Figure 1. Optical detection of glutamate release at individual IHC synapses: low and variable voltage threshold.**

A Maximum intensity projections of the organ of Corti from the right ear of a P17 mouse injected at P6 with 1–2 μl of AAV9.*hSyn*.iGluSnFR suspension, immunolabeled for iGluSnFR (GFP), IHCs, OHCs, and SGNs (parvalbumin) and synaptic ribbons and nucleus (CtBP2/RIBEYE). Close-up (right panel) highlights IHCs innervated by several iGluSnFR-expressing SGN boutons. Step sizes are 0.6 (left) and 0.5 μm (right), respectively. (Scale bars: 10 μm).

B Simultaneous perforated patch-clamp and imaging of IHCs (1.3 mM $[Ca^{2+}]_e$). IHCs were patch-clamped from the pillar side, with a patch pipette containing TAMRA-conjugated CtBP2-binding peptide. Toward the end of the recording, the membrane patch was ruptured to fill the IHC with the peptide, which stains synaptic ribbons and nucleus. (Left) Two exemplary confocal sections of IHCs showing baseline fluorescence of iGluSnFR-expressing afferent boutons from both modiolar (IHC1) and pillar (IHC2) side of the cell. Glutamate release from IHCs was evoked upon step depolarizations and detected as fluorescence change (ΔF) of iGluSnFR signal located on the SGN membrane (See (B′)). (Right) Overlaid images of the IHC1 and IHC2, displaying the boutons (iGluSnFR) and the synaptic ribbons (CtBP2), after the recording. (*: transduced afferent boutons, >: synaptic ribbons; Scale bar: 2 μm; see Appendix Fig S2 for the iGluSnFR-ROI detection routine).

B′ The stimulus protocol of the example IHCs from (B), displaying the voltage stimulation ($V_{stim}$), whole-cell $Ca^{2+}$ influx ($I_{Ca}$) and single-synapse iGluSnFR responses. IHCs were stimulated by 10-ms-long step depolarizations to −22 mV from the holding potential of −87 mV (1.3 mM $[Ca^{2+}]_e$), and iGluSnFR fluorescence was recorded at 50 Hz.

C Average $ΔF/F_0$ iGluSnFR traces in response to 10-ms-long step depolarizations from the holding potential (−87 mV) to a voltage within the physiologically relevant range of receptor potentials: from −62 to −22 mV (applied in pseudo-randomized order, step-size 5 mV, perforated patch-clamp, 1.3 mM $[Ca^{2+}]_e$, n = 29 boutons, N = 11 IHCs from nine mice). Shaded areas show ± SD.

D The voltage threshold of glutamate release was low and variable (−47.11 ± 4.09 mV, mean ± SD). The area under the curve of the iGluSnFR signal (top; $AUC(ΔF/F_0)_{10ms}$) from (C) and corresponding whole-cell $Q_{Ca}$ (bottom; mean ± SD). Detectable signals were defined here if the peak iGluSnFR signal was two times higher than baseline SD (depicted with *). All synapses had detectable signals in response to depolarizations ≥ −42 mV. (See also Figs EV1 and EV2).

and in the whole recorded voltage range (from −57 to 23 mV; Pearson's $r = 0.58$, $P < 0.0001$, Pearson's $r = 0.61$, $P < 0.0001$, respectively). These findings corroborated our observation in Fig EV1F and G that iGluSnFR saturation is not a major concern under these conditions.

Next, we compared SV pool dynamics by synaptic iGluSnFR-AUC versus whole-cell $ΔC_m$. To estimate the dynamics of RRP and sustained exocytosis, we fitted the sum of an exponential and a linear function. The resulting time constants of RRP depletion were 11.39 ms for iGluSnFR-AUC and 7.94 ms for $ΔC_m$ (Fig EV1E and H, and see Materials and Methods). By estimating an average AZ RRP of ~ 10 SVs from $ΔC_m$ measurements, we obtained an average change in iGluSnFR-AUC (0.23 arbitrary units) per SV. The sustained component of exocytosis amounted to 42.7 a.u./s (~ 185 SV/s per AZ) for the recorded synapses compared to 242 fF/s (504 SV/s per AZ; Fig EV1H). The faster rate derived from whole-cell $ΔC_m$ measurements likely reflects the contribution of extrasynaptic exocytosis (Pangrsic *et al*, 2015). Hence, ΔF-iGluSnFR can detect smaller amounts of exocytosis than whole-cell $ΔC_m$ and, on average, reports similar SV pool dynamics for single AZs.

Finally, we studied how the RRP is recruited by brief graded depolarizations in the range of physiological receptor potentials (Russell & Sellick, 1983). Using near-physiological conditions (perforated patch configuration, 1.3 mM $[Ca^{2+}]_e$), we applied 10-ms-long step depolarizations from the holding potential (−87 mV) to −62 mV in 5 mV increments up to −22 mV (applied in pseudo-randomized order; Figs 1C and D, and EV2A and B). We analyzed the voltage dependence of glutamate release by fitting Boltzmann functions to iGluSnFR-AUC (See Materials and Methods; Fig EV2C and D). The voltage threshold ($V_{10}$, defined as potential eliciting 10% of the maximum response) for glutamate release on average was −47.11 mV with considerable variance (SD: 4.09 mV; Fig EV2C and D). This is close to the threshold of activation of $Ca_V1.3$ $Ca^{2+}$ channels (~ −60 mV) (Zampini *et al*, 2010) and the *in vivo* resting membrane potential of IHCs (~ −55 mV) (Johnson *et al*, 2011). The variable voltage thresholds for glutamate release among the individual IHC synapses hint to differences in their transfer functions.

## Sequential dual-color imaging of $Ca^{2+}$ signal and glutamate release at single active zones

How the opening of $Ca_V1.3$ $Ca^{2+}$ channels translates into glutamate release critically shapes synaptic transfer and is determined by the topography of $Ca^{2+}$ channels and SV release sites. Previous studies evaluating the summed activity of several AZs indicate that a few channels in nanoscale proximity govern the $[Ca^{2+}]$ driving fusion of individual SVs at mature hair cell synapses ($Ca^{2+}$ nanodomain-like control of exocytosis) (Brandt *et al*, 2005; Graydon *et al*, 2011; Wong *et al*, 2014; Pangrsic *et al*, 2015). In the $Ca^{2+}$ nanodomain-like control of release, a near-linear relation between release and $Ca^{2+}$ influx is expected (apparent $Ca^{2+}$ cooperativity of release ($m$) close to 1) when varying the number of open $Ca^{2+}$ channels. Here, we found a near-linear dependence (operationally defined as $m < 2$) of glutamate release at single AZs on the whole-cell $Ca^{2+}$ influx when depolarizing IHCs within the range of physiological receptor potentials (Fig EV3A–C, $m_{QCa} = 1.55$, $n = 29$ synapses, $N = 11$ IHCs). Varying the voltage in this hyperpolarized range changes the open-channel number, and for $Ca^{2+}$ nanodomain control, this is more relevant than the change in single-channel current as once the channel opens the ensuing $Ca^{2+}$ signal tends to saturate the $Ca^{2+}$ sensor of release (Pangrsic *et al*, 2015). In contrast, and consistent with the supralinear intrinsic $Ca^{2+}$ dependence of exocytosis in IHCs (Beutner *et al*, 2001; Wong *et al*, 2014), reducing the effective single-channel current by the rapid flicker-block of $Ca^{2+}$ channels by $Zn^{2+}$ showed $m > 2$ (Fig EV3D–F, $m_{QCa} = 2.56$, $n = 24$ synapses, $N = 10$ IHCs). Taken together, imaging of glutamate release as a function of whole-cell $Ca^{2+}$ influx corroborates the notion of a $Ca^{2+}$ nanodomain-like control of release in mature IHCs (Brandt *et al*, 2005; Goutman & Glowatzki, 2007; Wong *et al*, 2014; Pangrsic *et al*, 2015). However, the presynaptic heterogeneity of $Ca^{2+}$ signaling in IHCs (Frank *et al*, 2009; Ohn *et al*, 2016) underscores the need of studying $Ca^{2+}$ influx–release coupling at individual AZs.

Studying the $Ca^{2+}$ dependence of exocytosis at individual AZs has remained difficult. Previous studies related whole-cell $Ca^{2+}$ influx to either whole-cell exocytosis (Brandt *et al*, 2005; Wong *et al*, 2014; Pangrsic *et al*, 2015) or to postsynaptic recordings of SV release (Goutman & Glowatzki, 2007). However, IHC synapses

fundamentally vary in voltage dependence, number, and clustering of their Ca²⁺ channels (Frank *et al*, 2009; Meyer *et al*, 2009; Ohn *et al*, 2016; Neef *et al*, 2018). Here, we studied the operation of individual IHC synapses in the physiologically relevant voltage range

during the 4ᵗʰ postnatal week (P21–26). We combined ruptured patch-clamp of IHCs with sequential dual-color imaging of first synaptic Ca²⁺ signals and then glutamate release (Fig 2A). To image synaptic Ca²⁺ signals, we used the red-shifted, low-affinity (Kd:19 µM)

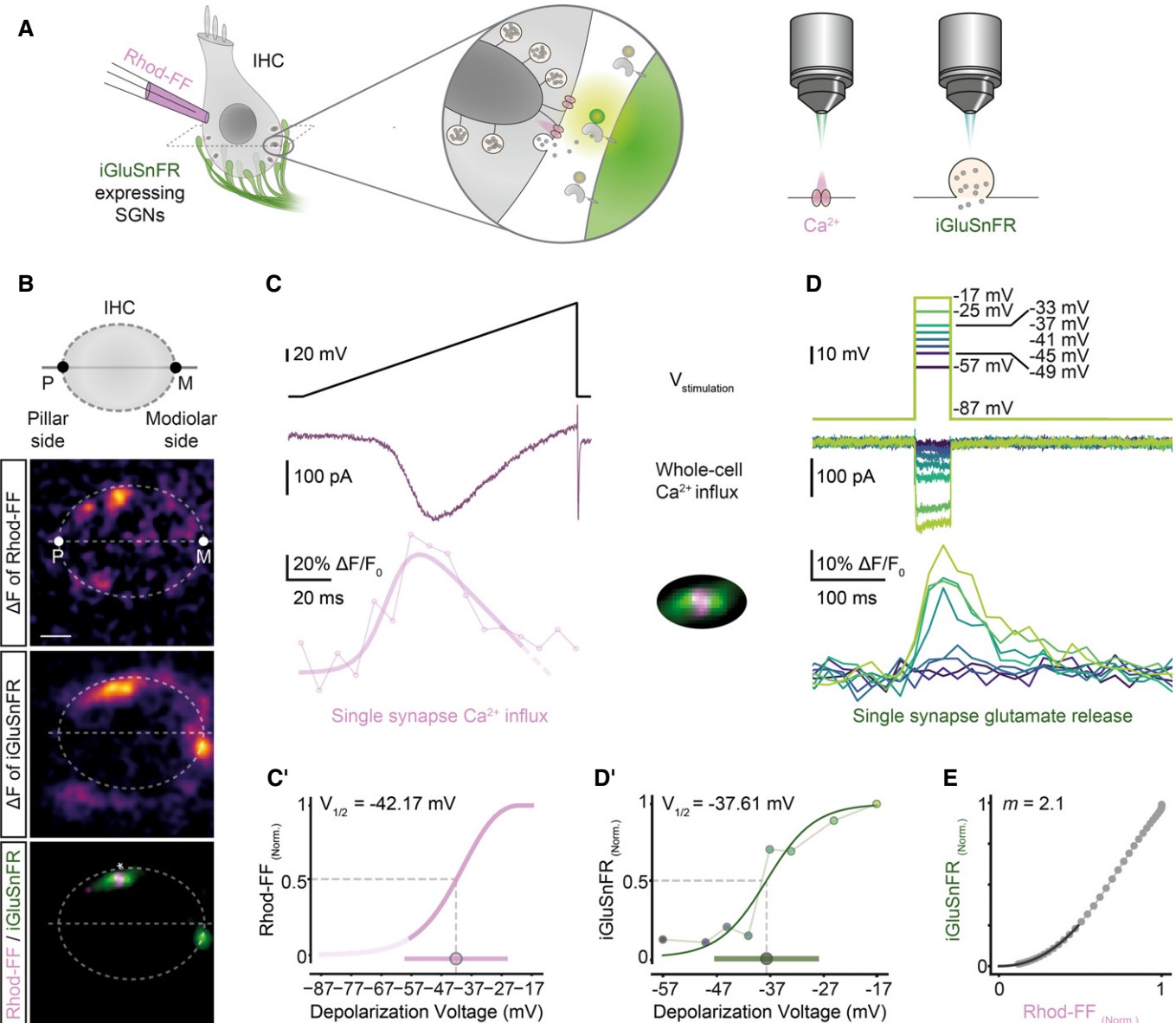

**Figure 2.  Sequential dual-color imaging of synaptic Ca²⁺ influx and glutamate release.**

A   IHCs were patch-clamped in ruptured-patch configuration with 800 µM Rhod-FF in the patch pipette, and simultaneously imaged for Ca²⁺ signals or glutamate release by spinning disk microscopy.

B   Mean ΔF images of Rhod-FF and iGluSnFR signals in response to a voltage ramp and a 50-ms-long step depolarization, respectively. The synapse marked with * on the overlaid image is further analyzed in the following panels. (P: pillar side, M: modiolar side; Scale bar: 2 µm)

C, D   Voltage command (top), corresponding whole-cell Ca²⁺ influx (middle) and the functional fluorescence responses (bottom; band-stop filtered at 33.3 Hz) from Rhod-FF and iGluSnFR, respectively. A modified Boltzmann function (see Materials and Methods, $R^2 = 0.81$) was fitted to a Rhod-FF fluorescence trace in response to a voltage ramp (C, C'). iGluSnFR-AUC was calculated per depolarization voltage and used for a Boltzmann fit (D', $R^2 = 0.92$). The voltage of half-maximal activation ($V_{1/2}$) and the dynamic range (10–90%) of synaptic Ca²⁺-influx and glutamate release were calculated from the fits and depicted as circle and bar in (C' and D'), respectively.

E   The obtained fits from the Ca²⁺ "hot spot" (C') and from glutamate release (D') were plotted against each other in a voltage range from −57 to −17 mV in 1 mV increments. A power function was fitted up to the 25% of the maximum iGluSnFR-AUC ($R^2 = 0.99$) to obtain the *m* estimate. (ruptured patch-clamp, 10 mM intracellular EGTA, 5 mM [Ca²⁺]ₑ; See also Fig EV4).

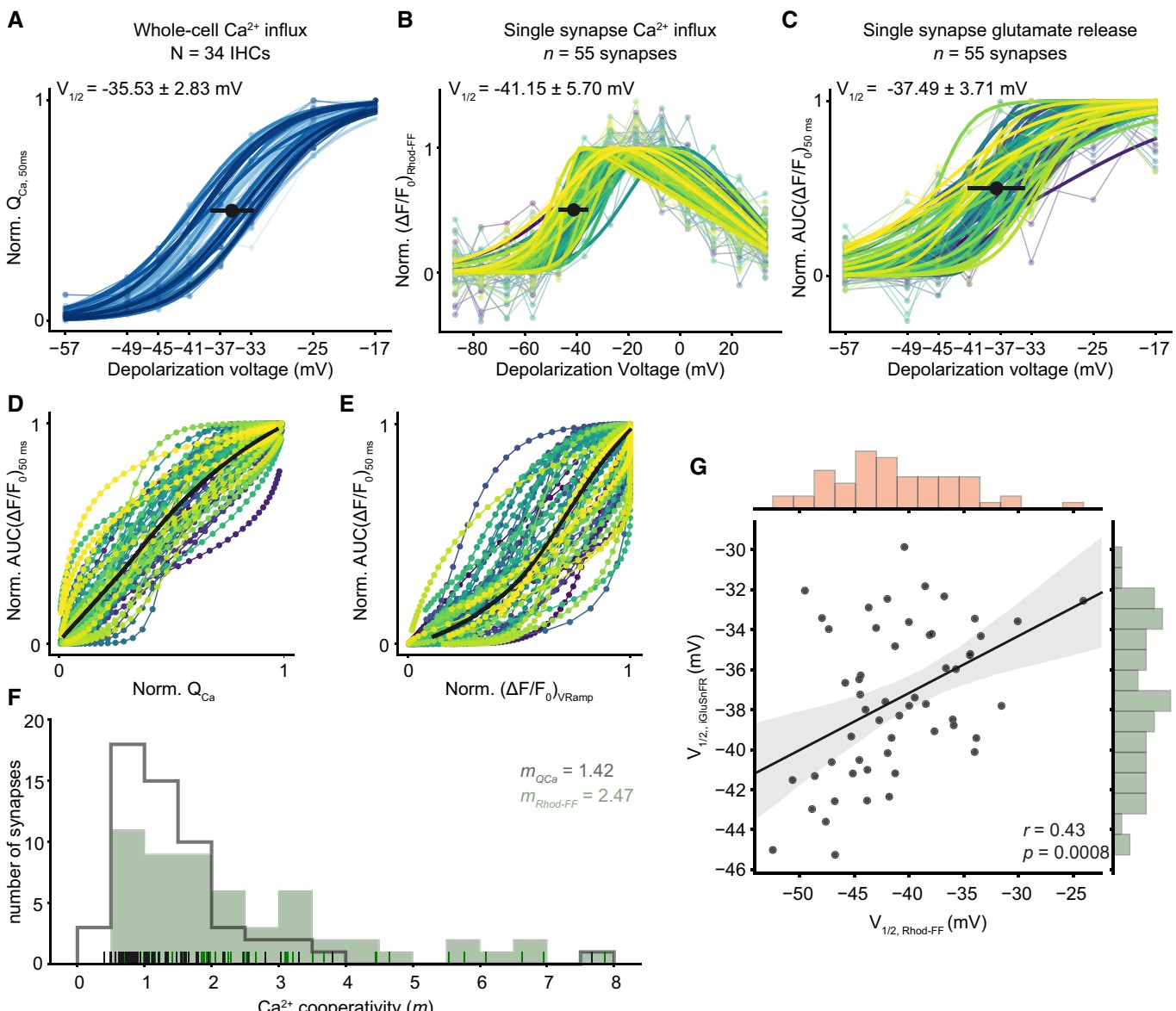

**Figure 3. IHC synapses vary in voltage dependence and apparent Ca²⁺ dependence of release.**

A  Normalized whole-cell $Q_{Ca}$, calculated in response to 50-ms-long step depolarizations, is plotted as a function of depolarization voltage. A Boltzmann function was fitted to estimate the $V_{1/2}$. Individual IHCs are color coded in shades of blue (mean ± SD, N = 34 IHCs from 28 mice, ruptured patch-clamp, 10 mM intracellular EGTA, 5 mM $[Ca^{2+}]_e$).

B  A voltage ramp was applied to obtain ΔF of Rhod-FF as a proxy of synaptic Ca²⁺ influx. The $V_{1/2}$ of synaptic Ca²⁺ influx was calculated from a modified Boltzmann function (see Materials and Methods) fitted to ΔF/F₀. (mean ± SD, n = 55 synapses; individual synapses are color coded; see Appendix Fig S7 for individual fits).

C  Normalized iGluSnFR-AUC, in response to 50-ms-long step depolarizations, same as (A) (see Appendix Fig S8 for individual fits).

D  The relation of whole-cell Ca²⁺-influx (A) and the synaptic glutamate release (C).

E  The relation of synaptic Ca²⁺ influx (B) and glutamate release (C). The bold lines show the means. (See also Appendix Fig S9 for individual plots)

F  The histogram shows the Ca²⁺ cooperativities (m) obtained by individual power function fitting until 25% of normalized iGluSnFR response from ((D); gray) and from ((E); green). The mean m was found to be 1.42 and 2.47, respectively. The rug plot shows the individual data points.

G  The $V_{1/2}$ of synaptic Ca²⁺ influx and glutamate release is correlated (Pearson's r = 0.43, P = 0.0008, Student's t-test). The marginal histograms show the distribution of each axis. Linear regression analysis (solid lines) and the associated 95% confidence intervals (shaded area). (See also Figs EV3–EV5).

chemical Ca²⁺-indicator Rhod-FF. We isolated Ca²⁺ signals at individual AZs using strong cytosolic buffering (10 mM EGTA in the patch pipette) and increased $[Ca^{2+}]_e$ (5 mM) (Frank *et al*, 2009; Ohn *et al*, 2016; Neef *et al*, 2018). We recorded glutamate release

using ΔF-iGluSnFR in SGN boutons contacting the mid-section of IHC.

Next, we studied the voltage dependence of synaptic Ca²⁺ influx and glutamate release. To probe the voltage dependence of synaptic

$Ca^{2+}$ influx, we imaged Rhod-FF fluorescence (Fig 2B and C) while applying a voltage ramp (−87 to +63 mV), a fast protocol inducing a minimum cellular $Ca^{2+}$ load (Ohn *et al*, 2016). This allowed us to analyze the $Ca^{2+}$ influx of the AZ corresponding to a given SGN bouton, by repeating the voltage ramps in five different planes (separated by 0.5 μm). The hot spots of Rhod-FF fluorescence elicited by depolarizations localized to the plasma membrane (Fig 2B) and to the synaptic ribbon (Fig EV4), indicating a cytosolic rise of $[Ca^{2+}]$ near the $Ca^{2+}$ channel cluster of the AZ. Then, to probe the voltage dependence of glutamate release, we imaged iGluSnFR (in the central plane of $Ca^{2+}$ imaging), while applying 50-ms-long step depolarizations (ranging from −57 to −17 mV) in a pseudo-randomized fashion (Fig 2B and D). We used 50-ms-long depolarizations to elicit sufficient IHC release in the presence of 10 mM EGTA, which is expected to constrain the $Ca^{2+}$ signal to the nanometer proximity of the $Ca^{2+}$ channels and to slow $Ca^{2+}$-dependent SV replenishment (Moser & Beutner, 2000). Furthermore, the iGluSnFR-AUC seemed robust toward saturation up to at least 100 ms of stimulation (see Fig EV1D–G and Appendix Fig S6). We analyzed the voltage dependences of synaptic $Ca^{2+}$ influx and glutamate release by fitting Boltzmann functions to $\Delta F/F_0$ of Rhod-FF (modified Boltzmann function, see Materials and Methods) and iGluSnFR-AUC. In the example shown in Fig 2, the voltages of half-maximal activation ($V_{1/2}$) of $Ca^{2+}$-influx (Fig 2C′) and glutamate release (Fig 2D′) were −42.2 and −37.6 mV, respectively. The resulting fit functions were then used to relate the glutamate release and the synaptic $Ca^{2+}$ influx over the physiologically relevant voltage range. To estimate the apparent $Ca^{2+}$ dependence of release for individual synapses, we fitted a power function on this relation (Fig 2E). We restricted the fit until the 25% of maximum iGluSnFR-AUC for all synapses, in order to avoid saturation, e.g., due to RRP depletion and obtained an $m$ estimate ($m = 2.1$ for the exemplary synapse). In conclusion, the sequential dual-color imaging approach allowed us to study $Ca^{2+}$ dependence of release at individual AZs.

## IHC synapses vary in voltage dependence and apparent $Ca^{2+}$ dependence of release

When systematically analyzing IHCs for the voltage dependence of whole-cell $Ca^{2+}$ influx ($Q_{Ca}$), synaptic $Ca^{2+}$ influx ($\Delta F/F_0$ of Rhod-FF), and glutamate release (iGluSnFR-AUC), we observed fundamental heterogeneity of AZs within and across IHCs. The threshold for glutamate release was −48.27 ± 6.47 mV (SD, $n = 55$ synapses, $N = 34$ IHCs from 28 mice) and showed a broader distribution than that of the whole-cell $Ca^{2+}$ influx ($V_{10} = −47.16 ± 3.19$ mV, SD, $P < 0.0001$, Levene's test). As previously reported (Ohn *et al*, 2016), synaptic $Ca^{2+}$ influx ($V_{1/2} = −41.15 ± 5.70$ mV, SD; $n = 55$ synapses, $N = 34$ IHCs from 28 mice, Fig 3B, see Appendix Fig S7 for individual fits) showed a broader and more negative $V_{1/2}$ distribution than that of the whole-cell $Ca^{2+}$ influx ($V_{1/2} = −35.44 ± 2.83$ mV, SD, Fig 3A). The $V_{1/2}$ distribution of synaptic glutamate release ranged from −45.25 to −29.86 mV and also showed a more negative average $V_{1/2}$ (−37.49 ± 3.71 mV, SD, $n = 55$ synapses, $P = 0.008$, Fig 3C, see Appendix Fig S8 for individual fits) than the whole-cell $Ca^{2+}$ influx. The $V_{1/2}$ distributions differed significantly between glutamate release and synaptic $Ca^{2+}$ influx (Fig EV5 and $P = 0.009$, Levene's test) as well as between synaptic and whole-cell $Ca^{2+}$ influx ($P = 0.001$, Levene's test). The

$V_{1/2}$ values of synaptic $Ca^{2+}$ influx and glutamate release correlated (Pearson's $r = 0.43$, $P = 0.0008$, Fig 3G). This correlation indicates that the differences in the voltage dependence of $Ca^{2+}$ influx are propagated to release, generating heterogeneous output among the IHC AZs at a given receptor potential.

Next, we estimated the apparent $Ca^{2+}$ dependence of release by relating the iGluSnFR-AUC to the whole-cell $Ca^{2+}$ influx (Fig 3D) and to the synaptic $Ca^{2+}$ influx (Fig 3E) for the above voltage protocol that primarily varied the open-channel number (see also Figs 1C and D, and EV3). The apparent $Ca^{2+}$ dependence of glutamate release at the single-synapse level (Fig 3F) was higher on average ($m_{Rhod-FF} = 2.47$, see Appendix Fig S9 for the data of individual IHCs) than when relating glutamate release to the whole-cell $Ca^{2+}$ influx ($m_{QCa} = 1.42$). The observed discrepancy of the average $m$ estimates from single-synapse analysis and the $m$ estimated based on the whole-cell $Q_{Ca}$ might at least partially be explained by the exclusion of synapses at the densely innervated basal IHC cap. We suspect that there are a fair number of synapses with $m < 2$ in the basal cap. The $m$ estimate based on the whole-cell $Q_{Ca}$ was lower on average than the $m$ estimate obtained in the perforated-patch configuration on P15–19 IHCs at 1.3 mM $[Ca^{2+}]_e$ using a similar stimulus protocol (see Figs 1C and D, and EV3A and C, $P = 0.001$, Mann–Whitney *U*-test). This difference is compatible with the hypothesis that strong $Ca^{2+}$ buffering by 10 mM EGTA favors the operation of release sites under $Ca^{2+}$ nanodomain control. Since the data of Fig EV3 were acquired at an earlier developmental stage (P15–19), the developmental tightening of the $Ca^{2+}$ channel–exocytosis coupling (Wong *et al*, 2014) might also have contributed to the lower $m$ in Fig 3D (P21–26). We did not find a significant difference between the thresholds of release with physiological buffering (perforated patch-clamp, 1.3 mM $[Ca^{2+}]_e$) and with high EGTA buffering (ruptured patch-clamp, 10 mM intracellular EGTA, 5 mM $[Ca^{2+}]_e$, Mann–Whitney *U*-test, $P = 0.42$), supporting the lack of obvious glutamate release quench in these high buffering conditions. Nevertheless, probing the synaptic $Ca^{2+}$ dependence of release in more physiological buffering conditions remains important task for future studies. When operationally defining $m < 2$ ("near-linear") as indicative of $Ca^{2+}$ nanodomain-like control of release, we found approximately half of the synapses (29 out of 55 synapses) to operate in this scenario. The other half showed a broad spread of $m$ values reaching up to 8, compatible with $Ca^{2+}$ microdomain-like control of release despite the presence of 10 mM EGTA. Taken together, single-synapse imaging of $Ca^{2+}$ influx and glutamate release revealed a heterogeneity of the apparent $Ca^{2+}$ dependence of release that is likely due to differences in the coupling of $Ca^{2+}$ channels to release sites among the IHC AZs. This heterogeneous coupling of $Ca^{2+}$ channels to release sites likely contributes to the heterogeneous output of IHC AZs.

## Pillar synapses operate at more negative potentials than modiolar synapses

SGNs exhibit a spatial preference in their IHC innervation pattern in the cat: High SR–low threshold fibers innervate the pillar side of the IHC, and low SR–high threshold fibers contact the modiolar side of the IHC (Liberman, 1982). Assuming analogy for mouse IHCs, we probed how presynaptic heterogeneity might contribute to the diversity of SGNs by analyzing the synaptic $Ca^{2+}$ influx and glutamate

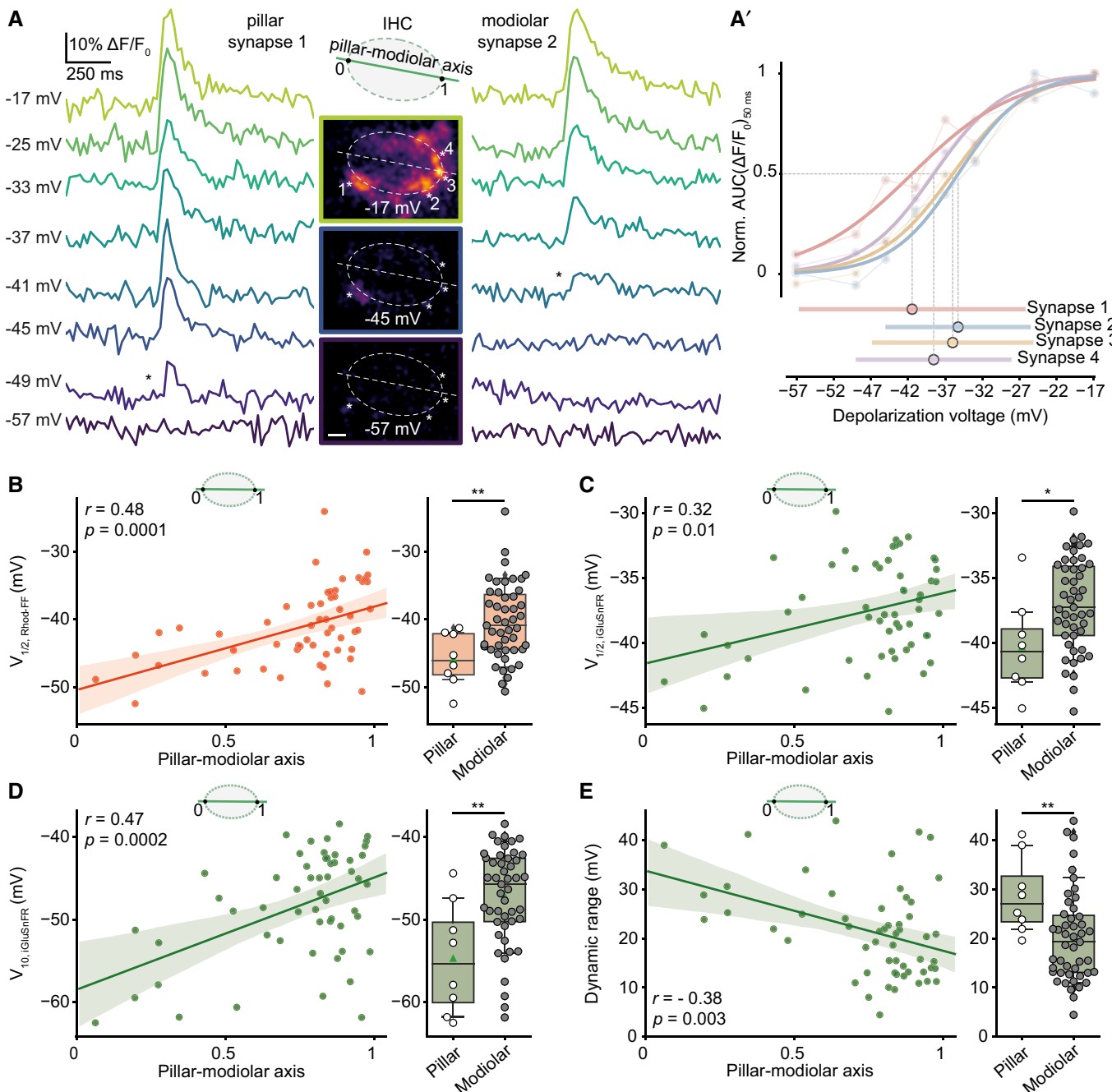

**Figure 4. Pillar synapses are active at more negative potentials than modiolar synapses.**

A The single cell example shows release dynamics of two synapses innervating the same IHC from either pillar or modiolar side. iGluSnFR signals (band-stop filtered at 33.3 Hz) in response to 50-ms-long step depolarizations to the given depolarization voltages are depicted (ruptured patch-clamp, 10 mM intracellular EGTA, 5 mM $[Ca^{2+}]_e$). The middle panel shows $\Delta F$ images of iGluSnFR, recorded from the mid-section of the IHC. * depict the first detected response in the given synapse. Note that pillar synapse 1 is already active at −49 mV, while modiolar synapse 2 starts responding only at −41 mV. (Scale bar: 2 μm).

A' The normalized iGluSnFR-AUC as a function of depolarization voltage. Dynamic range and $V_{1/2}$ of the synapses are depicted in the lower panel.

B–E Left panel shows the linear regression analysis (solid lines) of $V_{1/2}$ of synaptic $Ca^{2+}$ influx (B), $V^{1/2}$ (C), threshold (D), and dynamic range (E) of glutamate release as a function of position along the pillar–modiolar axis. Shaded areas depict the associated 95% confidence intervals. Significance of correlation coefficients is reported by a two-tailed *P*-value. Right panel shows box and whisker plots of these properties of synapses grouped into pillar and modiolar halves of the IHCs. Box plots indicate first quartile (25th percentile), median and third quartile (75th percentile) with whiskers reaching from 10 to 90%. For comparison of pillar and modiolar synapses, either Student's *t*-test (for normally distributed data) or Mann–Whitney *U*-test (for non-normally distributed data) was applied. *$P \leq 0.05$, **$P \leq 0.01$.

release as a function of position along the pillar–modiolar axis. Figure 4A compares the simultaneous iGluSnFR responses of two exemplary synapses of one IHC: One positioned at the pillar and the other one at the modiolar side. In this example, the pillar synapse had a lower threshold; i.e., it already became active at −49 mV, while the modiolar synapse only started to respond at −41 mV. The pillar synapse also showed a more negative $V_{1/2}$ and a wider dynamic range compared to the modiolar synapses in the given cell (Fig 4A′). As this observation was made in the same section of an IHC and was representative for the population (Fig 4B–E), we consider potential technical reasons unlikely. On average, $V_{1/2}$ of both $Ca^{2+}$ influx and glutamate release, and threshold of glutamate release (Fig 4B–D) were more negative at pillar synapses. Moreover, pillar synapses showed a wider dynamic range of release (Fig 4E). Nonetheless, there was substantial variability in particular among the modiolar synapses. This is obvious from the example shown in Fig 4A′ as well as at the population level (Fig 4B–E). We also performed linear regression analysis on the relation of pillar–modiolar position and $V_{1/2}$ of synaptic $Ca^{2+}$ influx as well as glutamate release (Fig 4B–E). This analysis confirmed a pillar–modiolar gradient of $V_{1/2}$ of synaptic $Ca^{2+}$ influx ($r = 0.48$; $P = 0.0001$) that we previously reported for an earlier postnatal stage (P14–20) (Ohn et al, 2016). Furthermore, it showed a pillar–modiolar gradient of the voltage threshold ($r = 0.47$; $p = 0.0002$), $V_{1/2}$ ($r = 0.32$; $P = 0.013$), and dynamic range ($r = −0.38$; $P = 0.003$) of release. $Ca^{2+}$ cooperativity did not show a significant correlation with the position along the pillar–modiolar axis ($r = 0.17$; $P = 0.21$; Appendix Fig S10). However, $Ca^{2+}$ cooperativities ≥ 3 were mainly found for AZs on the modiolar side.

In conclusion, pillar and modiolar AZs differed in their voltage dependence and dynamic range of glutamate release. These differences between pillar and modiolar synapses support the hypothesis that diversity of the spontaneous and sound-evoked SGN firing is, at least in part, rooted in the heterogeneous biophysical properties of presynaptic $Ca^{2+}$ channels and transmitter release. Activation at more negative potentials of pillar synapses agrees with the high SR and low sound threshold of the corresponding SGNs. However, the wider dynamic range of glutamate release at pillar AZs is more difficult to reconcile with the narrower dynamic range of sound encoding of high SR–low threshold SGNs (Taberner & Liberman, 2005). While the exciting hypothesis of a modiolar–pillar segregation of synaptic properties has kept instructing important experiments, we should be aware of bias arising from this model. In fact, the present data seem to support a model that is between a strict ordering and a "salt-and-pepper" distribution of AZ properties in IHCs.

### Clustering of synaptic properties indicates three synapse subtypes likely distinguished by their implementation of $Ca^{2+}$ influx–release coupling

By the sequential dual-color imaging of synaptic $Ca^{2+}$ influx and release, we quantified each synapse with 15 parameters. Correlations among several of these parameters can be intuitively explained, such as the voltage dependence of a synapse, which is primarily rooted in that of $Ca^{2+}$ channel activation (Fig EV6). Moreover, a supralinear coupling of release to $Ca^{2+}$ influx is expected to compress the dynamic range of release. While the comparative

analysis of pillar and modiolar synapses (see Fig 4) aims to elucidate synaptic correlates of the functional SGN properties, it has both value and limits. For an unbiased and in-depth analysis, we applied K-means clustering ($K = 3$) on the $11^{th}$-dimensional space of single-synapse properties (excluding positional and whole-cell information) and obtained three synapse clusters (putative subtypes; see Appendix Fig S11 for clustering results with $K = 2$ and $K = 4$). To visualize the clusters in two or three dimensions, we performed principal component analysis (PCA) and used the first three principal components that explained 79% of the variance (Fig 5A–D). The main factors contributing to the first three PCs were the dynamic range of glutamate release, thresholds of glutamate release, and synaptic $Ca^{2+}$ influx (Appendix Fig S12).

Next, we checked the response properties of these putative synapse subtypes. The first subtype of synapses showed a low $m$ ($1.56 \pm 1.07$, 27 synapses) indicating a $Ca^{2+}$ nanodomain-like control of release as well as a wide dynamic range ($27.52 \pm 9.37$ mV) and a hyperpolarized activation threshold ($−53 \pm 5.07$ mV) of release. In contrast, the third synapse subtype showed a high $m$ ($6.10 \pm 1.18$, six synapses) with smaller dynamic range ($15.53 \pm 3.95$ mV) and a more depolarized threshold ($−42 \pm 2.8$ mV; Fig 5F–I). Synapses from the first cluster are predicted to be active already around the IHC receptor potential ($−55$ mV, Fig 5I) (Johnson et al, 2011). The properties of the second synapse subtype fell in between those of subtypes 1 and 3. Finally, we evaluated positions of the three synapse subtypes in an effort to match them to the concept of a SGN-subtype-specific innervation pattern along the pillar–modiolar axis. Pillar synapses were exclusively of subtype 1, while modiolar synapses were distributed across all subtypes. This is exemplified by three neighboring modiolar synapses of an IHC (Fig 5E): Each synapse belongs to a separate cluster, employing different apparent $Ca^{2+}$ dependencies (see Appendix Fig S9 for the data of all individual IHCs). Hence, different topographies of $Ca^{2+}$ channels and SV release sites apparently coexist even in compact presynaptic cells. How the IHC manages to vary the AZ organization and, thereby, $Ca^{2+}$ influx–exocytosis coupling remains to be elucidated. Our data indicate some extent of pillar–modiolar segregation of synaptic properties, while substantial heterogeneity is found among the AZ at all positions. Taken together, we propose a model in which IHCs fractionate the coding of sound intensity information, contained in the receptor potential, via heterogeneous synaptic input–output functions that are sourced by differences in voltage dependence and release site coupling to $Ca^{2+}$ channels (Fig 5J).

## Discussion

The auditory system encodes sound pressures over a range of six orders of magnitude. The sensory mechanisms contributing to this fascinating behavior include active cochlear micromechanics, adaptation at various stages, as well as diversity of SGNs and their synapses with IHCs. At each tonotopic position in the cochlea, SGNs differ in their spontaneous and evoked firing, effectively tiling the range of audible sound pressures with their responses (Kiang et al, 1965; Sachs & Abbas, 1974; Liberman, 1978; Winter et al, 1990; Taberner & Liberman, 2005). Differences in the transcriptomic profiles of SGNs have led to their categorization into molecular

subtypes (Sun *et al*, 2018; Shrestha *et al*, 2018; Petitpré *et al*, 2018). Moreover, heterogeneity in the pre- and postsynaptic properties of afferent IHC-SGN synapses as well as of efferent SGN innervation have been proposed to explain the fractionation of the entire range of audible sound pressures, detected by the same IHC, into different neural representations (Ruel *et al*, 2001; Frank *et al*, 2009; Liberman

*et al*, 2011; Ohn *et al*, 2016; Neef *et al*, 2018). Here, we studied the transfer at individual IHC-SGN synapses, by imaging of synaptic Ca²⁺ signals and glutamate release during IHC patch-clamp recordings. On average, glutamate release had a voltage threshold near the physiological resting potential (Figs 1 and 4) and showed a near-linear dependence on the whole-cell Ca²⁺ influx when primarily

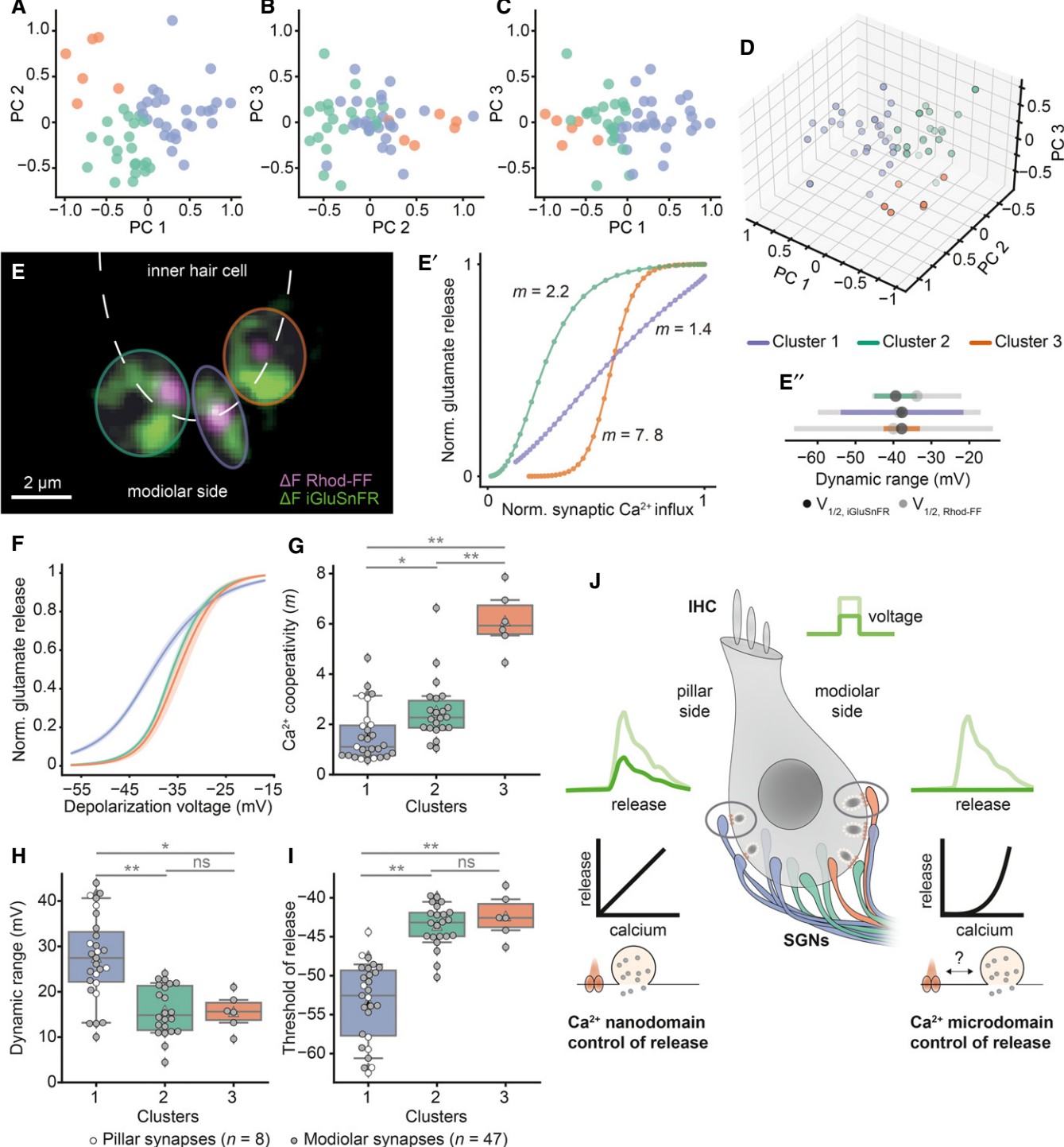

**Figure 5.**

◄

**Figure 5. Three putative synapse subtypes, co-existing in individual IHCs, differ in their synaptic transfer functions and $Ca^{2+}$ dependencies.**

11 synaptic properties (except for the positional and whole-cell information; see Fig EV6) of 55 synapses ($N = 34$ IHCs from 28 mice, ruptured patch-clamp, 10 mM intracellular EGTA, 5 mM $[Ca^{2+}]_e$) were used for the PCA and $K$-means clustering.

A–C    The 2D plots of the first three principal components (PCs), labeled based on the clusters obtained by $K$-means clustering.
D        3D plot showing the first three PCs.
E        Single IHC exhibits different modes of $Ca^{2+}$-control of release. The overlaid mean $\Delta F$ image of Rhod-FF (magenta) and iGluSnFR (green) shows synaptic $Ca^{2+}$ influx and glutamate release at three neighboring modiolar synapses (individual synapses are color coded based on their clusters).
E′       The relation between $Ca^{2+}$ influx and glutamate release of given synapses in the voltage range of $-57$ to $-17$ mV plotted with 1 mV increments. A power function was fitted until the 25% of normalized iGluSnFR-AUC. Synapses showed different $Ca^{2+}$ dependencies ($m = 2.2, 1.4, 7.8$; green, purple, orange).
E″      Dynamic ranges of corresponding $Ca^{2+}$ influx (gray) and glutamate release (color coded) with $V_{1/2}$ depicted. Note that the synapses from Cluster 1 (purple), employing $Ca^{2+}$ nanodomain-like control of release ($m = 1.4$), exhibit wider dynamic range than the other synapses.
F–I     Mean glutamate release (iGluSnFR-AUC) as a function of depolarization voltage in the three identified clusters (mean ± SEM, Cluster 1: $n = 27$ synapses, Cluster 2: $n = 22$ synapses, Cluster 3: $n = 6$ synapses). The $Ca^{2+}$ cooperativity ($m$) (G), dynamic range (H), and glutamate release threshold ($V_{10}$) (I) show differences for the three clusters. Cluster 1 is composed of linear synapses with wider dynamic range and lower threshold compared to the other clusters. Pillar synapses are depicted as white-filled circles, and modiolar ones as gray-filled circles. Box plots indicate first quartile (25th percentile), median and third quartile (75th percentile) with whiskers reaching from 10–90%. The clusters were compared by one-way ANOVA test, followed by a *post hoc* Tukey's test. *$P \leq 0.05$, **$P \leq 0.01$.
J         The proposed model for sound intensity encoding in an IHC. Differences in the presynaptic control of release, in terms of $Ca^{2+}$ signaling and $Ca^{2+}$ channel–exocytosis coupling, enable IHC to diversify the transfer functions of individual synapses for the same receptor potentials.

varying the open-channel number (Fig EV3). We then employed sequential dual-color imaging of $Ca^{2+}$ signals and glutamate release of single AZs (Fig 2). This way we revealed the heterogeneity of synaptic transfer functions, rooted in differences in voltage dependence of the $Ca^{2+}$ influx and in its coupling to exocytosis (Fig 3). We found a pillar–modiolar gradient of AZ transfer functions: consistent with the preferred innervation of high-SR SGNs on the pillar side, pillar AZs operated at more negative potentials than the modiolar ones (Fig 4). By $K$-means clustering of AZs according to the single-synaptic parameters, we obtained three putative synapse subtypes, differing in the voltage dependence, apparent $Ca^{2+}$ dependence, and dynamics of release (Fig 5).

**Heterogeneity of $Ca^{2+}$ channel–release coupling**

Studying $Ca^{2+}$ signals and glutamate release at individual AZs revealed that the coupling of $Ca^{2+}$ influx to exocytosis varies among the AZs, unlike previously assumed based on average AZ behavior (Brandt *et al*, 2005; Goutman & Glowatzki, 2007; Wong *et al*, 2014; Pangrsic *et al*, 2015). We show that it ranges from $Ca^{2+}$ nanodomain to $Ca^{2+}$ microdomain control of exocytosis and differs even among neighboring IHC AZs. Approximately one-half of the AZs showed $Ca^{2+}$ nanodomain-like control of exocytosis (operationally defined as $m < 2$) during primary manipulation of the open-channel number. Hence, different from what was proposed based on an AZ summation model (Heil & Neubauer, 2010), $Ca^{2+}$ nanodomain-like control of exocytosis occurs at individual IHC AZs. However, we found the other half of the AZs with $m > 2$ suggesting $Ca^{2+}$ microdomain control of exocytosis to co-exist. We assume all AZs share the same intrinsic $Ca^{2+}$ cooperativity of 4–5 that was established for IHC exocytosis by $C_m$ recordings (Beutner *et al*, 2001; Brandt *et al*, 2005). This likely reflects the cooperative binding of 4–5 $Ca^{2+}$ ions to the putative $Ca^{2+}$ sensor of IHC release, otoferlin (Roux *et al*, 2006). In the 4th postnatal week, we consider IHCs to have reached maturity but further changes might occur (Liberman & Liberman, 2016). However, the heterogeneity was preserved in all recorded ages (P21–26, Appendix Fig S13). Here, we studied synapses of apico-cochlear IHCs and, given the previously described tonotopic differences in $Ca^{2+}$ channel–release coupling on the whole-cell level

(Johnson *et al*, 2017), single-synapse analysis in basocochlear IHCs remains an important task for the future. Taken together, we propose a model by which IHCs vary the topographies of $Ca^{2+}$ channels and SV release sites at the AZs likely to diversify synaptic transmission beyond the heterogeneity of AZ size and $Ca^{2+}$ signaling.

Non-exclusive candidate mechanisms for position-dependent AZ diversification in IHCs include (i) developmental competition of synapses, (ii) transsynaptic signaling from SGNs, and (iii) planar polarity signaling. One interesting idea is that, during development, pioneer SGN axons (Druckenbrod et al, 2020) making the first synaptic contact at the modiolar side of IHCs attract large amounts of presynaptic resources. In addition to such potential transsynaptic cueing of AZ properties, IHCs might employ their planar polarity signaling to instruct modiolar–pillar gradients of AZ size (Jean *et al*, 2019). How such signaling differentially shapes AZ properties remains an exciting topic for future studies. Clearly, polarized trafficking of components of the $Ca^{2+}$ channel complex or other AZ proteins could contribute. This might, for instance, involve the adapter and PDZ-domain protein Gipc3, defective in human deafness (Charizopoulou *et al*, 2011), required for the modiolar–pillar gradient of maximal synaptic $Ca^{2+}$ influx (Ohn *et al*, 2016). Intriguingly, the spatial distribution of the number of $Ca^{2+}$ channels and their voltage dependence might be regulated by different mechanisms (Jean *et al*, 2019). While the number of $Ca^{2+}$ channels scales with AZ/ribbon size (Frank *et al*, 2009; Ohn *et al*, 2016; Neef *et al*, 2018) and shows a modiolar–pillar gradient, the topography relative to SV release sites and biophysical properties of the $Ca^{2+}$ channels seem to follow an opposing gradient. For instance, the voltage dependence of synaptic $Ca^{2+}$ influx (Ohn *et al*, 2016; Jean *et al*, 2019, and Fig 4) and, consecutively, glutamate release (Fig 4) shows a pillar–modiolar gradient, with activation at more hyperpolarized potential for the pillar AZs that show smaller $Ca^{2+}$ channel clusters. Likewise, $m$ of exocytosis was typically lower at the pillar side, indicating that $Ca^{2+}$ nanodomain control of release prevails with the lower number of $Ca^{2+}$ channels, while an increased channel number favors domain overlap as also predicted by modeling (Wong *et al*, 2014). Future studies will need to probe for differences in the abundance of potential molecular linkers (Han *et al*, 2011; Liu *et al*, 2011; Jung *et al*, 2015; Krinner *et al*, 2017) or spacers of $Ca^{2+}$

channels and release sites such as RIM and RIM-BP among the IHC AZs and to explore actual $Ca^{2+}$ channels–release site topography of IHC AZs e.g. by electron microscopy (Nakamura *et al*, 2015).

As for size and $Ca^{2+}$ channel number of the AZ, the positional dependence of biophysical properties and topography of $Ca^{2+}$ channels might be shaped during postnatal development. For instance, AZs of immature IHCs, on average, employ a $Ca^{2+}$ microdomain-like control of exocytosis. Maturation tightens the coupling of $Ca^{2+}$ influx and exocytosis at least when considering the collective behavior of all IHC AZs (Wong *et al*, 2014), coinciding with the appearance of high-SR fibers (Wong *et al*, 2013). Our study indicates that a subset of mature IHC synapses employs $Ca^{2+}$ microdomain control of release, representing an additional mechanism employed by the IHC to diversify synaptic transmission and endowing this subset with further potential of presynaptic plasticity (Vyleta & Jonas, 2014).

Potential pitfalls of our imaging approach to single IHC AZ function include the following: (i) glutamate spill-over from neighboring synapses, (ii) saturation of iGluSnFR and other non-linear effects resulting from the glutamate binding to endogenous receptors in the postsynapse, and (iii) contamination of synaptic $Ca^{2+}$ signals by mitochondrial-$Ca^{2+}$ changes, as some rhodamine-based dyes partition into mitochondria (Oheim *et al*, 2014). We aimed to minimize synaptic crosstalk by choosing well-separated boutons. Moreover, simulations showed iGluSnFR provides a linear indication of glutamate release (Armbruster *et al*, 2020). While we cannot rule out a contribution of mitochondrial Rhod-FF, the use of 10 mM EGTA would make mitochondrial-$Ca^{2+}$ uptake unlikely. This is supported with the strict co-localization of the Rhod-FF signals with the synaptic ribbon (Fig EV4).

### Relating presynaptic heterogeneity to functional SGN diversity

The functional subtypes of SGNs are said to spatially segregate their IHC innervation: High-SR SGNs preferentially innervate the pillar side of the IHC, while low-SR SGNs preferentially innervate the modiolar side of the IHC in the cat cochlea (Liberman, 1982). Here, we demonstrate by iGluSnFR imaging in the apical organ of Corti of mice that glutamate release from pillar synapses operates at more hyperpolarized potentials than the modiolar ones. This offers an exciting presynaptic hypothesis for the functional SGN diversity: Resting potential or weak receptor potentials will primarily recruit pillar synapses, which can readily explain the high SR and low sound threshold of SGNs innervating the pillar IHC side. While this hypothesis was previously phrased based on the heterogeneous voltage dependence of $Ca^{2+}$ influx (Ohn *et al*, 2016), whether and how this translates into heterogeneity of glutamate release remained

unclear. Indeed, unlike we had previously assumed (Ohn *et al*, 2016), the present study revealed differences in the $Ca^{2+}$ channel–release coupling among the IHC AZs.

Heterogeneity of $Ca^{2+}$ channel–release coupling, too, could contribute to diversifying SGN function. First, $Ca^{2+}$ nanodomain control of exocytosis would increase the SR, as stochastic opening of $Ca^{2+}$ channels could trigger release in such a tight coupling scenario (Moser *et al*, 2006; Eggermann *et al*, 2012). Interestingly, AZs of vestibular type I hair cells on average show very tight coupling of $Ca^{2+}$ channels and exocytosis (Vincent *et al*, 2014) and are innervated by neurons with high spontaneous activity (~ 90 spikes/s) (Goldberg & Fernandez, 1971). Second, $Ca^{2+}$ nanodomain control of exocytosis likely also promotes low voltage threshold for the same reason, and indeed, we found a correlation between the threshold of release and $m$ (Fig EV6). Third, $Ca^{2+}$ nanodomain control of exocytosis widens the dynamic range of release. Indeed, we found a negative correlation between dynamic range and $m$ (Fig EV6). One of the most puzzling questions is how the pillar AZs with more hyperpolarized operation driving high-SR SGNs that show a smaller dynamic range of sound encoding can be reconciled with the wider dynamic range of glutamate release found for pillar AZs. One possible explanation is high SR at the *in vivo* IHC resting potential lowers the standing RRP available for sound-evoked release and hence causes a narrower dynamic range of release (reviewed in ref. (Moser *et al*, 2019)). Indeed, disruption of PDZ protein Gipc3 resulted in enhanced SR and narrower dynamic range (Ohn *et al*, 2016). Furthermore, as rate-level functions are usually assessed in response to sound stimuli lasting more than 50 ms, there is likely a contribution of RRP replenishment (Pangrsic *et al*, 2015). We cannot exclude possible contribution of postsynaptic properties, efferent modulation, dynamic range adaptation (Wen *et al*, 2009), and non-linearity imposed by the basilar membrane (Sachs & Abbas, 1974) on the dynamic range of SGN firing. Lastly, differences in $Ca^{2+}$ channel–exocytosis coupling could affect the short-term plasticity (reviewed in ref. (Böhme *et al*, 2018)), note the higher adaptation strength of high-SR SGNs *in vivo* (Heil & Peterson, 2015). A similar example of linear and non-linear synapses comes from the zebrafish bipolar cells (Odermatt *et al*, 2012) that encode light intensities over four orders of magnitude. In conclusion, we propose a model where differences among the IHC AZs in the presynaptic control of release, in terms of presynaptic $Ca^{2+}$ signaling and $Ca^{2+}$ channel–exocytosis coupling, enable a single IHC to diversify the synaptic signaling to SGNs for the same receptor potential. We suggest that non-linear transformation of the sensory signal by heterogeneous synapses extends the dynamic range of intensity coding.

## Materials and Methods

### Reagents and Tools table

| Reagent/resource | Reference or source | Identifier or catalog number |
|---|---|---|
| **Experimental Models** | | |
| C57BL/6J (*Mus musculus*) | | |

                                                    

**Reagents and Tools table** (continued)

| Reagent/resource | Reference or source | Identifier or catalog number |
|---|---|---|
| **Recombinant DNA** | | |
| pAAV-hSyn-iGluSnFR | Marvin *et al* (2013) | RRID:Addgene_98929 |
| **Antibodies** | | |
| Chicken anti-GFP | Abcam | Abcam Cat# ab13970, RRID:AB_300798 |
| Guinea pig anti-parvalbumin | Synaptic Systems | Cat# 195 004, RRID:AB_2156476 |
| Mouse anti-CtBP2 | BD Biosciences | Cat# 612044, RRID:AB_399431 |
| **Chemicals, enzymes, and other reagents** | | |
| Rhod-FF, tripotassium salt | AAT Bioquest, Biomol | Cat #: 21075 |
| TAMRA-conjugated CtBP2 peptide | Zenisek *et al* (2004) | BioSynthan |
| Abberior Star 488-conjugated CtBP2 peptide | Zenisek *et al* (2004) | BioSynthan, MPIbpc |
| Amphotericin B | Merck (Calbiochem) | Cat #: 171375 |
| **Software** | | |
| Python | Python Software Foundation | https://www.python.org/psf/ |
| Igor | WaveMetrics | https://www.wavemetrics.com/ |
| PatchMaster (Software), HEKA EPC-10 (Hardware) | HEKA Electronic GmbH | https://www.heka.com/ |
| ImageJ | | https://imagej.nih.gov/ij/ |

## Methods and Protocols

### Animals and postnatal injections

All experiments were done in compliance with national animal care guidelines and were approved by the University of Göttingen Board for Animal Welfare and the Animal Welfare Office of the State of Lower Saxony (permit number: 17-2394). Postnatal AAV injections were made into scala tympani of the right ear through the round window (Akil *et al*, 2012; Jung *et al*, 2015). P5–7 WT C57Bl/6 mice were used for the injection of AAV9 virus under human synapsin promoter (pAAV9.*hSyn*.iGluSnFR.WPRE.SV40, Cat#98929-AAV9, Addgene, USA, or produced in our own laboratory, see Reagents and Tools Table, and Huet and Rankovic, 2021) to drive transgenic expression of iGluSnFR in SGNs. In brief, the right ear was accessed through a dorsal incision. Once the round window membrane was located, a quartz capillary pipette was used to gently puncture it and inject ~ 1–1.5 μl of pAAV9.*hSyn*.iGluSnFR (titer ≥ $1 \times 10^{13}$ vg/ml). Subsequently the wound was sutured. The whole procedure was performed under general isoflurane anesthesia. For analgesia, buprenorphine (0.1 mg/kg) was injected subcutaneously prior the surgery, additional local analgesia (xylocaine) was applied to the skin and post-surgery pain was covered by metamizol (1.33 mg/kg) provided in the drinking water for 5 days. The recovery of the animals was monitored on a daily basis. All animals were kept in a 12-h light/dark cycle, with access to food and water *ad libitum* and together with the mother until the end of the weaning period (~ P21). Injected WT mice were used for experiments either 1 week (P15–19) or 2 weeks (P21–26) after the injection.

### Auditory brainstem recordings

Recordings of auditory brainstem responses (ABR) were performed on P29 mice as previously described (Strenzke *et al*, 2016). Briefly, mice were anesthetized with a combination of ketamine (125 mg/kg) and xylazine (2.5 mg/kg) intraperitoneally. The core temperature was maintained constant at 37°C using a heat blanket (Hugo Sachs Elektronik–Harvard Apparatus). The TDT II system run by BioSig software (Tucker Davis Technologies) or by MATLAB (MathWorks) was used for stimulus generation, presentation, and data acquisition. Tone bursts (6/12/24 kHz, 10-ms plateau, 1 ms $\cos^2$ rise/fall) or clicks of 0.03 ms were presented at 40 Hz (tone bursts) or 20 Hz (clicks) in the free field ipsilaterally using a JBL 2402 speaker.

### Immunohistochemistry and confocal microscopy

The cochlea was fixed in formaldehyde (4% in phosphate-buffered saline [PBS], 1 h on ice). For immunostaining of the whole-mount organs of Corti, the apical turns of organs of Corti were dissected out and washed three times with PBS. For immunostaining of the cryosections, the cochlea was decalcified in 0.5 M EDTA overnight. After a PBS washing step, the cochleae were incubated in 25% sucrose in PBS at 4°C. The cochlea was frozen in Tissue-Tek and cryosectioned with section thickness of 16 μm.

The samples (cochlear apical turns or cryosections) were blocked with a goat serum dilution buffer (16% normal goat serum, 450 mM NaCl, 0.3% Triton X-100, 20 mM phosphate buffer, pH 7.4) for 1 h at room temperature in a wet chamber. The blocking was followed by an overnight incubation with the primary antibodies at 4°C. After 3 × 5 min PBS washing steps, the samples were incubated with the secondary antibodies for 1 h at room temperature. Following the final 4 × 5 min PBS washing steps, the samples were mounted in mounting medium (Mowiol 4-88, Sigma). The primary antibodies used were the following: mouse anti-CtBP2 (1:200, BD Biosciences, 612044, See Reagents and Tools table)—to detect synaptic ribbons —chicken anti-GFP (1:200, Abcam, 13970)—to detect iGluSnFR— and guinea pig anti-parvalbumin (1:200, Synaptic Systems, 195004) —to detect SGNs, OHCs, and IHCs. Secondary goat antibodies were used with 1:200 dilution: Alexa Fluor 488-conjugated anti-chicken (Dianova, 703-45-155), Alexa Fluor 633-conjugated anti-mouse

(Invitrogen, A31571), Alexa Fluor 488 anti-chicken (Invitrogen, A11039), and Alexa Fluor 568 anti-guinea pig (Invitrogen, A11075). Images were acquired using an Abberior Instruments Expert Line STED microscope, with excitation lasers at 488, 561, and 640 nm using a 1.4 NA 100× or 20× oil immersion objective, in confocal mode. Z-step sizes of 0.5 or 0.6 μm were used with 100× or 20× objectives, respectively. Images were adjusted for brightness and contrast using ImageJ for illustration purposes.

### Patch-clamp recordings

The apical 2/3 turn of organs of Corti was acutely dissected from P15 to P26 animals in HEPES Hanks' solution containing (in mM): 5.36 KCl, 141.7 NaCl, 10 HEPES, 0.5 $MgSO_4$, 1 $MgCl_2$, 5.6 D-glucose, and 3.4 L-glutamine (pH 7.2, ~ 300 mOsm/l). The IHC basolateral membranes were exposed by cleaning of nearby cells with a suction pipette by approaching from either pillar or modiolar side. All experiments were conducted at room temperature (20–25°C). Patch pipettes were made from GB150-8P or GB150F-8P borosilicate glass capillaries (Science Products, Hofheim, Germany) for perforated and ruptured patch-clamp recordings, respectively. To decrease the capacitive noise, pipettes were coated with Sylgard and their tips were polished with a custom-made microforge. All patch-clamp recordings were done simultaneously with fluorescent imaging of iGluSnFR or of iGluSnFR and $Ca^{2+}$.

### Perforated patch recordings

Perforated patch-clamp was performed as described previously (Moser & Beutner, 2000). For $Ca^{2+}$ current and membrane capacitance ($C_m$) measurements, the extracellular solution contained the following (in mM): 110 NaCl, 35 TEA-Cl, 2.8 KCl, 1 $MgCl_2$, 1 CsCl, 10 HEPES, 1.3 $CaCl_2$, and 11.1 D-glucose (pH 7.2, ~ 305 mOsm/l) and was introduced into the recording chamber via a perfusion system. The pipette solution contained (in mM): 130 Cs-gluconate, 10 TEA-Cl, 10 4-AP, 10 HEPES, 1 $MgCl_2$, as well as 300 mg/ml amphotericin B (pH 7.17, ~ 290 mOsm/l). The intracellular solution also contained the TAMRA-conjugated dimeric CtBP2/RIBEYE-binding dimer peptide (10 μM, Biosynthan, Germany, See Reagents and Tools table) (Francis *et al*, 2011; Wong *et al*, 2014). To label the synaptic ribbons, the peptide was introduced to the cell by rupturing the membrane patch toward the end of the recordings. The intracellular exposure to amphotericin (used to perforate the membrane for electrical access) did not result in a noticeable increase in IHC conductance and compromise IHC health. All the measurements were done via EPC-10 amplifiers controlled by Patchmaster software (HEKA Elektronik, Germany). The holding potential was −87 for all the recordings. All voltages were corrected for liquid junction potential offline (17 mV). Currents were leak-corrected using a p/10 protocol. Recordings were used only if the leak current was lower than 30 pA and the series resistance (Rs) was lower than 30 mOhm. The Lindau-Neher technique was used to measure the $C_m$ changes (Lindau & Neher, 1988). Exocytosis was quantified from $C_m$ changes as described previously (Moser & Beutner, 2000; Neef *et al*, 2014). IHCs were stimulated by step depolarizations of different durations (2–100 ms, applied in a pseudo-randomized manner) to −23 mV at intervals of 60–100 s (Fig EV1D–G). To probe voltage dependence of release, IHCs were step depolarized for 10 ms to potentials ranging from −62 to −22 mV in 5 mV

increments in a pseudo-randomized order (Figs 1 and EV2). For the $Zn^{2+}$ perfusion experiments, 1 mM $Zn^{2+}$ was slowly perfused in and out of the recording chamber, while IHCs were step depolarized for 10 ms to −23 mV simultaneously up to 20 times (Figs EV1A and B, and EV3D-F).

### Ruptured patch recordings

Ruptured patch experiments were performed in extracellular solution containing (in mM): 2.8 KCl, 102 NaCl, 10 HEPES, 1 $CsCl_2$, 1 $MgCl_2$, 35 TEA-Cl, 2 mg/ml D-Glucose, and 5 $CaCl_2$ (pH 7.2, 300 mOsm). The patch pipette solution contained (in mM): 111 L-glutamate, 1 $MgCl_2$, 1 $CaCl_2$, 10 EGTA, 13 TEA-Cl, 20 HEPES, 4 Mg-ATP, 0.3 Na-GTP and 1 L-Glutathione (pH 7.3, ~ 290 mOsm). For fluorescent imaging, 800 μM of the low-affinity chemical $Ca^{2+}$ indicator Rhod-FF tripotassium salt (Kd:19 μM, AAT Bioquest, USA, See Reagents and Tools table) was added to the intracellular solution. The recordings were discarded when the leak current exceeded −50 pA at −87 mV or $R_S$ was greater than 15 MΩ within 4 min after break-in. For $Ca^{2+}$ imaging experiments, a voltage ramp (from −87 to +63 mV during 150 ms; 1 mV/ms) was applied to evoke $Ca^{2+}$ influx (Figs 2 and 3).

For all IV recordings, the IHCs were step depolarized for 20 ms from −87 to +63 mV in 5 mV increments. The IV recordings were used to assess the fitness of the cell, and recordings were discarded when the $Ca^{2+}$ current rundown exceeded 25%.

### Spinning disk confocal imaging of $Ca^{2+}$ and iGluSnFR

Imaging experiments were performed with a spinning disk confocal scanner (CSU22, Yokogawa, Germany) mounted on an upright microscope (Axio Examiner, Zeiss, Germany) with 63×, 1.0 NA objective (W Plan-Apochromat, Zeiss). The spinning disk speed was set to 2,000 rpm to avoid uneven illumination. A scientific CMOS camera (Neo, Andor, Northern Ireland) with a pixel size of 103 nm was used to acquire images. iGluSnFR and Rhod-FF or TAMRA peptide were excited by diode-pumped solid-state lasers with 491 nm and 561 nm wavelength, respectively (Cobolt AB).

### iGluSnFR imaging

To avoid photobleaching, iGluSnFR-expressing SGN boutons were detected via low intensity 491 nm excitation. The imaging plane for the target IHC was selected when several transduced boutons were visible, in the mid-basal section of the cell, avoiding the high synapse density at the basal pole. A brief step depolarization was applied to the cell to check for the functional signal in the given plane. iGluSnFR fluorescence was acquired at 50 Hz simultaneously with patch-clamp recordings. The iGluSnFR signal was evoked by step depolarizations of different durations to different voltage values, as it is specified in every dataset.

### Sequential dual-color imaging of $Ca^{2+}$ and iGluSnFR

For the sequential dual-color imaging of $Ca^{2+}$ and glutamate release, as described above for iGluSnFR imaging, the imaging plane was selected based on the baseline fluorescence of iGluSnFR. Once the middle plane was set, the fluorescence of Rhod-FF was imaged at 100 Hz while $Ca^{2+}$ currents were triggered by applying five voltage ramps (from −87 to +63 mV, 1 mV/ms) in five alternating planes separated by 0.5 μm. To precisely control the Z-plane, a piezo positioner for the objective (Piezosystem, Germany) was used. After the

Ca$^{2+}$ imaging, the iGluSnFR signal was acquired at 50 Hz by applying 50-ms-long step depolarizations from the holding potential of −87 to different voltage values. Depolarizations (to −57, −49, −45, −41, −37, −33, −25, −17 mV) were applied in a pseudo-randomized manner and covered the dynamic range of IHC glutamate release.

### Data analysis
#### Patch-clamp recordings
Electrophysiological recordings were analyzed using custom-written programs in Igor Pro 6.3. Whole-cell Ca$^{2+}$ charge ($Q_{Ca}$) was calculated by the time integral of the leak-subtracted current during the depolarization step. $\Delta C_m$ was calculated as the difference between the average $C_m$ 400 ms before and after the depolarization, skipping the initial 100 ms after the depolarization.

#### Imaging of iGluSnFR
Image and further data analysis and visualization were done in Python (Python software foundation) with custom written code using the following Python libraries: NumPy, Pandas, Matplotlib, Skimage, SciPy, Sklearn, statsmodels, and Seaborn.

**Region of interest detection** The $\Delta F$ image was created by subtracting baseline fluorescence ($F_0$, an average of 15 frames before stimulus) from the fluorescence images acquired during/after stimulation (F, an average of five frames). The $\Delta F$ image was median-filtered with a two-dimensional pixel array size of 4–6 depending on the signal amplitude. To create a mask for ROI detection, maximum entropy thresholding was applied to the median-filtered $\Delta F$ image. To label and separate individual ROIs, a watershed segmentation algorithm was used. A single mask was generated per cell, using the recording with strongest stimulation, and applied for all images. Individual ROIs, corresponding to postsynaptic SGN boutons, were further confirmed by the nearby presence of presynaptic ribbon peptide (TAMRA-conjugated dimeric CtBP2-binding peptide). The fluorescence of every pixel in the defined ROI was averaged over time for further analysis. The background fluorescence was calculated by averaging 60 × 60 pixels in the pillar region of the image, where no iGluSnFR fluorescence is expected: By their anatomy, SGNs innervate IHCs and leave the cochlea toward the modiolus.

**Analysis of fluorescence traces** The average background value was subtracted from the raw fluorescence traces (F). Following background subtraction, $\Delta F$ traces were generated by subtraction of mean baseline fluorescence ($F_0$). $\Delta F$ was normalized to $F_0$ to create $\Delta F/F_0$ traces. For peak detection, $\Delta F/F_0$ traces were smoothened using a Hanning window function with a window size of 7. To correct for photobleaching, we fitted a single exponential to $\Delta F/F_0$ traces. The area under the curve (AUC) was estimated by calculating the area between $\Delta F/F_0$ and the fit in an interval of 40 frames from the beginning of the stimulus.

**Estimation of sensitivity of iGluSnFR and $\Delta C_m$** For iGluSnFR, mean of 10 frames before stimulation is compared pairwise per synapse with the mean of four frames after stimulation. For $\Delta C_m$, mean of 400 points before and after stimulation is used for pairwise comparison per cell.

**Estimation of the time to peak and decay constant** To obtain decay constant ($\tau_{off}$), the following function was fitted to the 30 points of photobleaching corrected $\Delta F/F_0$ traces after stimulation.

$$\Delta F/F_0 = A * (1 - e^{-t/\tau_{on}}) * e^{-t/\tau_{off}}$$

$$\mathrm{timetopeak(ms)} = \tau_{on} * \log(\tau_{off}/\tau_{on} + 1)$$

**Estimation of the RRP size and time constant of RRP depletion** To assess the dynamics of RRP and sustained exocytosis, we fitted a sum of an exponential and linear function (Pangrsic *et al*, 2015) to $\Delta Cm$ and iGluSnFR-AUC for different stimulus durations.

$$\Delta C_m(t)\, or\, \mathrm{AUC}_{iGluSnFR}(t) = \mathrm{RRPsize} * (1 - e^{t/\tau})^n + \mathrm{slope} * t.$$

With the assumptions of $\Delta C_m$ of ~ 40 aF contributed by a single SV (Grabner & Moser, 2018) and ~ 12 AZs (Meyer *et al*, 2009) for the apical IHCs, we obtained iGluSnFR-AUC increase of 0.23 a.u. per SV.

#### Sequential imaging of Ca$^{2+}$ and iGluSnFR

**ROI detection–iGluSnFR** The ROIs were picked as described above. Differently, a Gaussian filter with sigma of 1–3 was applied consistent with the detection of Ca$^{2+}$ hot spots (see below). A mean mask was generated per cell using all the recordings. ROIs were confirmed by the presence of a corresponding Ca$^{2+}$ "hot spot".

**ROI detection–Rhod-FF** Similarly, a $\Delta F$ image was created from the mean time series, in this case, by averaging all the trials from five recorded planes. This $\Delta F$ image was treated the same way as described for iGluSnFR-ROI detection. The created mask was applied to all recording planes and the plane with the maximum $\Delta F$ for a given ROI was used for further analysis. This way we used the plane with the highest signal for each Ca$^{2+}$ "hot spot".

**Analysis of fluorescence traces** To remove the noise caused by the spinning disk speed at 2,000 rpm (33.3 Hz), obvious in the Fourier amplitude spectrum, the raw traces were filtered with a 33.3 Hz band-stop filter. The obtained traces were background-subtracted and normalized to $F_0$ as described above. Fluorescence–voltage (FV) relations for iGluSnFR were estimated from the step depolarizations to different voltage values. iGluSnFR-AUC for each depolarization was calculated as described above.

**Estimation of the parameters of threshold, dynamic range, and $V_{1/2}$** A Boltzmann fit was used to estimate the two fitting parameters: voltage of half-maximum activation ($V_{1/2}$) and slope factor ($k$) of glutamate release.

$$\mathrm{AUC}_{iGluSnFR}(V) = \frac{1}{1 + e^{\frac{V - V}{k}}}.$$

For Ca$^{2+}$ imaging, FV curves were estimated from voltage ramps. To optimize the raw FV traces against noise such as readout or shot noise from the CCD camera, the following equation was used (Ohn *et al*, 2016):

$$F_{Rhod-FF}(V) = F_0 + \frac{g_{max}(V - V_r)}{1 + e^{\frac{V_{1/2} - V}{k}}}.$$

The slope factor ($k$) was obtained with this equation. The resulting fit was used to estimate $V_{1/2}$ by minimizing the scalar at the mid-point. The reversal potential ($V_r$) was fixed to +47.6 mV after LJ potential (17 mV) correction. In addition, this fit was used to calculate the fractional activation curve, dividing it by the extrapolated linear fit to the decay of fluorescence. To estimate fractional $V_{1/2,Pactivation}$ and $k_{1/2,Pactivation}$ (see Fig EV6), an additional Boltzmann fitting was done.

The peak of the Rhod-FF signal was obtained by averaging three frames corresponding to the voltage values between −17 and 3 mV during ramp depolarization. Dynamic ranges were calculated from the fits as the voltage range of 10–90% of the maximal activation. Threshold was calculated as the voltage value where there is 10% of the maximal activation. Note that $V_{1/2}$ values obtained from fluorescence traces after denoising (band-stop filter at 33.3 Hz) were comparable to the ones from the raw fluorescence traces.

***Estimating the Ca²⁺ cooperativity*** FV fits for Rhod-FF or $Q_{Ca}$ and iGluSnFR-AUC were plotted against each other in the voltage range of −57 to −17 mV. To obtain single-synapse glutamate release–Ca²⁺ signal/whole-cell Ca²⁺ charge relationship, a power function was fitted:

$$AUC_{iGluSnFR}(V) = A(F_{Rhod-FF})^m$$

$$AUC_{iGluSnFR}(V) = A(Q_{Ca})^m.$$

***Calculation of the position of the synapse along cell's pillar–modiolar axis*** To estimate the cell boundary, an ellipse was fitted to the baseline fluorescence of Rhod-FF. We defined the pillar–modiolar axis of the cell as the major axis of the ellipse. We calculated the shortest distance from the center of a given Ca²⁺ "hot spot" to the normalized major axis. A number was assigned to a given synapse on the normalized scale from 0 (pillar side) to 1 (modiolar side).

***K-mean clustering and principal components analysis*** K-means clustering algorithm ($K = 3$) was applied to the whole dataset of 11 synaptic properties for three clusters (see Fig 5). Principal component analysis was used to display the clusters obtained by the K-means clustering.

***Statistical analysis***
All the statistical tests were performed in Python (Python Software Foundation). Averages are expressed as mean ± SD or as mean ± SEM (specified in the figure legends), and box plots indicate 25–75 quartile with whiskers reaching from 10 to 90%. Datasets were checked for normal distribution by D'Agostino & Pearson omnibus normality test and for equality of variances. For normally distributed data, unpaired two-tailed student's *t*-test was applied, and for non-normally distributed data, Mann–Whitney *U*-test was used. Dependent samples were compared by paired *t*-test (for normally distributed data) or Wilcoxon's signed rank test (for non-normally distributed data). Comparison of dispersion was performed by Levene's test. We used one-way ANOVA for multiple comparisons followed by *post hoc* Tukey's test. The Pearson correlation coefficient was used to test for linear correlation. Significant differences are reported as *$P \leq 0.05$, **$P \leq 0.01$, ***$P \leq 0.001$.

## Data availability

This study includes no data deposited in external repositories.

**Expanded View** for this article is available online.

## Acknowledgements

We thank Dres. Christian Vogl and Vladan Rankovic for their contributions in the initial phase of the project, N. Dietrich, S. Gerke, D. Gerke, and C. Senger-Freitag for expert technical assistance. We thank Dr. G. Ramos-Traslosheros for his initial assistance with the analysis. We thank Dr. Thomas Euler for providing us with a sample of AAV9 hSyn.iGluSnFR. We thank Dres. J. Neef, L.M. Jaime Tobón, E. Neher, and S.O. Rizzoli for helpful comments on the manuscript. We thank Dr. Vladimir Belov for providing Abberior Star 488-conjugated peptide. This work was supported by the Deutsche Forschungsgemeinschaft (DFG, German Research Foundation) under Germany's Excellence Strategy—EXC 2067/1-390729940 to T.M. as well as via the collaborative research center 889 and the collaborative research center 1286 to T.M. Open access funding enabled and organized by ProjektDEAL.

## Author contributions

TM and ÖDÖ designed the study. ÖDÖ performed the experiments and the analysis. TM and ÖDÖ prepared the manuscript.

## Conflict of interests

The authors declare that they have no conflict of interest.

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
