## [Review Process File · The EMBO Journal]

A sensory cell diversifies its output by varying Ca²⁺ influx-release coupling among active zones

Özge Demet Özçete and Tobias Moser
DOI: 10.15252/embj.2020106010

Corresponding author(s): Tobias Moser (tmoser@gwdg.de)

Review Timeline:

Submission Date:	23rd Jun 20
Editorial Decision:	16th Jul 20
Revision Received:	11th Oct 20
Editorial Decision:	27th Oct 20
Revision Received:	30th Oct 20
Accepted:	3rd Nov 20

Editor: Karin Dumstreif

Transaction Report:

Dear Tobias,

Thank you for submitting your manuscript to The EMBO Journal. Your manuscript has now been seen by two referees and their comments are provided below.

As you can see from the comments, the referees appreciate the introduced changes and support publication in The EMBO Journal. They raise a number of good points that would be good to address in a revised version. I am happy to discuss the raised points further and maybe it would be most helpful to do so via phone or video

Thank you for the opportunity to consider your work for publication. Looking forward to discussing the revisions further with you.

with best wishes

Karin

Karin Dumstrei, PhD
Senior Editor
The EMBO Journal

Further information is available in our Guide For Authors:

The revision must be submitted online within 90 days; please click on the link below to submit the revision online before 14th Oct 2020.

Referee #1:

In this remarkable study by Ozcete and Moser evidence is presented on how hair cells in mammalian ears are able to diversify their synaptic output to better encode sound information. In particular they address a long-standing question on how individual IHC are able to translate a wide range of sound intensities into complementary dynamic ranges in firing rates of SGN. The authors used a state of the art technique, with functional imaging methods to simultaneously track Ca²⁺ influx to IHC (with Rhod FF) and glutamate release (through the genetically encoded glutamate sensor iGluSnFR). They investigated in detail how the voltage dependence of different release sites within a given IHC differed. A correlation is informed, by which Ca²⁺ cooperativity was closer to one on pillar synapses, whereas activation threshold occurred at hyperpolarized potentials on this side of hair cells. Analysis is deep and informative. Conclusions on how these observations could explain high vs low SR fiber segregation are sound.

While the technique picked by the authors could not seem to be more appropriate, some methodological questions remain. These questions are laid out in the following paragraphs, with one that stands out are refers to the variability in the glutamate sensor signal. But also for the single synapse calcium. How variable are parameters like V_{1/2} or m when imaging the same spot multiple times? Have the authors evaluated this experiment? How variable is max DeltaF/F on a given glutamate spot? The authors outlined correlations, clearly showing variability for parameters like V_{1/2}, m, etc. But functional variability should be contrasted with variability due to detection constraints and/or trial-to-trial variance. Without this information some of the physiological

conclusions may be biased or, quite the opposite, loose strength.

Major points:

-Fig. 1. DF, why not DF/F0? Are those traces in Fig. 1B due to single IHC depolarization? Or average of several?

-Methods. For this reviewer it is not intuitive to represent the shape iGluSnFR ROIs. Particularly after applying this maximum entropy thresholding, and seeing images in Fig. 1B. From these representative images it seems as if iGluSnFR fluorescent spots have the size of an entire synaptic bouton (or even larger), whereas glutamate release sites are presumably much smaller. Do fluorescence transients occur over this large areas? How variable are they in pixel number? How compact are ROIs in shape? It would be highly advisable to illustrate the reader with a representative image of individual/average ROIs.

-Is focus set in Figures 2/3 similarly to Fig. 1?

-Saturation of iGluSnFR is naturally a problem but I do not think that a correlation between the fluorescent signal and C_m could reveal it. If the fluorescent protein could undergo saturation it would most likely occur early during an IHC depolarization, and comparing fluorescence and capacitance in long pulses (>50 ms) would not be necessarily informative. It should be more helpful to establish a correlation between these two variables with short IHC pulses in which the membrane potential was varied (e.g. between -47 and -22 mV).

-One important issue is variability/noise in the fluorescence signal. What's the variability in V1/2 of glutamate release (Fig. 3) when imaging multiple times the same spot? Same for the Ca^{2+} dependence of release. How variable is m when repeating the experiment over a single glutamate spot? I believe that a calibration of this variability is required to be certain about conclusions. From looking at individual curves on Figure 3 (although difficult to clearly see), it seems that neither the glutamate nor the calcium signals show monotonic trends but rather an erratic increase and decrease as membrane voltage changes. Could the authors refer to this?

-Similarly to the observation on how focus is set, one concerning issue is out of focus images in Figure 4. It is not clear from the narrative how many cases with multiple simultaneous glutamate spots were recorded by the authors, and/or are included in the dataset. But it may be problematic if within the dataset multiple glutamate spots were not imaged at the brightest focal plane. How would these properties differ if glutamate fluorescence was not in focus?

-How does m change (or not) with respect of the pillar/modiolar axis? Could the authors include a correlation with position as done in Fig. 4B-D?

-Why do the authors think that coupling between whole-cell calcium and glutamate fluorescence is rather linear, whereas this same relation between synaptic calcium and release seems on average supralinear (Fig. 3)? This point requires discussion and/or interpretation. Whole cell calcium could be considered a weighted average of synaptic influx, and so be the calculated ' m '. Shouldn't be this ' m ' equal to the average of single ' m 's calculated from individual synaptic sites?

Minor:

Line 33. ROI detection. Size of Median filtering, please indicate if size used is pixels.

Referee #2:

EMBOJ-2020-106010, corr. author Prof. Moser

"A sensory cell diversifies its output by varying Ca²⁺ influx-release coupling among active zones"

General summary and opinion about the principal significance of the study, its questions and findings:

Auditory inner hair cell (IHC) ribbon synapses have to deal with the great challenge of encoding an extremely wide range of sound intensities, a dynamic range of more than 100 dB. To perform this challenging task, auditory inner hair cells have partitioned their synaptic output sensitivity and dynamic range. Indeed, each IHC form synapses with a pool of 10 to 30 afferent nerve fibers, among which spontaneous activity, acoustic threshold and dynamic range vary widely. The present study tackles one of the most important questions in sensory neuroscience: what are the functional mechanisms that dictate the synaptic diversity of each type of sensory synapses? To address this question, the authors characterized the synaptic transfer function at each individual IHC synapse by imaging Ca²⁺-signals and glutamate release during IHC-patch clamp recording. The results showed that variation in the voltage-gating and spatial-organization of calcium channels at each presynaptic active zone could determine the firing specificity of the auditory nerve fibers.

Although the heterogeneity in amplitude and voltage-sensitivity of the calcium responses of the IHC active zones has already been extensively characterized in several previous papers from the same lab, this is the first report characterizing the synaptic transfer function correlating Ca²⁺ influx and glutamate release at each type of post-hearing ribbon synapses. The results nicely confirmed a pillar-modiolar gradient of AZ transfer functions: with pillar AZ, contacting low threshold high-SR fibers, being activated at more negative potentials and having a tighter spatial organization (nanodomain) between Ca²⁺ channels and release sites.

This study is technically sound and brings novel and interesting results for the general neuroscience community. The authors convincingly demonstrate how sensory synapses adapt their presynaptic AZ-Ca²⁺ signals to encode a large dynamic range of stimuli. The precise molecular mechanisms controlling this differential voltage gating and Ca²⁺ channel organization at each type of AZs remain now to be determined.

Specific major concerns essential to be addressed to support the conclusions: Most of my major questions concern the optical detection of glutamate release.

1. The fluorescent Glut responses from synaptic boutons were variable in time and amplitudes as shown in Fig 1B. Could these different responses be also explained by a variability in the kinetics of glut reuptake in each synaptic cleft, i.e. due to a differential expression of EAAT transporter between synapses? I think some comparative measurements should be performed in presence of TBOA to remove possible interference with differential Glut reuptake or buffering.

2. It is not clear whether all afferent nerve fibers were similarly labeled with the AAV expressing iGluSnFr; in other words, what is the percentage of labeled SGN or synaptic boutons? Also, is the labeling intensity, i.e. the expression level of the Glut sensor uniform between SGNs? I think this is an important issue because the peak-amplitude as well as the integral of the Glut fluorescent responses would greatly be dependent on a differential iGluSnFr density between afferent boutons? This point should be clearly addressed. Ideally, a comparative calibration of the Glut fluorescent responses between pillar and modiolar boutons using extracellular Glutamate-uncaging or Glutamate puffs at various concentrations should be tested.

3. Because of crowding, the Glut-responses were only measured on a small fraction of the synaptic boutons, those contacting apically the IHCs (as indicated page 5). It seems therefore nonrealistic to extrapolate the data in a model that considers only a small fraction of IHC synapses. Is it possible

that a classification would have fallen in more than 3 categories if all fibers would have been studied.

4. I wonder whether spontaneous release could be directly recorded by the glut sensor in IHC-contacting boutons, i.e. in absence of depolarizing and voltage-clamping IHCs? In this case, it would have been interesting to directly identify and compare high vs low Spont Release boutons.

5. The Pearson's r coefficient of correlation of the synaptic properties, ie for $V_{1/2}$ of Ca^{2+} influx and Glut release is quite low ($r < 0.5$; fig3G). At first glance of Fig 3G, it seems difficult to say that the synaptic factors are correlated. This point should be corrected and discussed. Also in Fig4B-E, the r coefficients are again very low (< 0.5). The conclusions drawn from these figures should be moderated and discussed.

6. It looks like that all ruptured WCR recordings were performed with 10 mM intracellular EGTA, thus quenching the Glut release responses of microdomain synapses. To clearly show a differential nanodomain vs microdomain organization for pillar and modiolar synapses, a direct comparison should be made with 1 mM intracellular EGTA in similar recording WCR conditions.

7. In Fig 6, 11 synaptic properties were used for the PCA and K-means clustering. Indicate which ones were selected because I see 15 synaptic properties in Fig 5. Why K was arbitrarily chosen or applied to be 3? I guess because of in vivo previously studies reported 3 categories of auditory nerve firing properties. How the data would behave with $K = 2$ or 4 for example? Could you define what are the three principal components PC1, PC2 and PC3 in Fig 6A-C. Please clarify the analysis of this complex Fig for the general reader.

Minor concerns that should be addressed:

- The title of the study seems too general. Only auditory IHC synapses were studied here; I suggest for example to replace sensory cell by "Auditory inner hair cells diversify their synaptic transfer function by....."
- Remove first sentence of the abstract that seems redundant with the next following sentence.
- The introduction and discussion are too long, sometimes repetitive and speculative. These sections should be shortened by a least one page.
- Indicate in text legend or methods, the values of the slice thickness of the confocal images. Also clearly indicate in figure 1B the locations of the measured ROI.
- Scale bar 10 μm of Fig 1B is too large - 2 μm is enough.
- Methods page 28: indicate the precise Cat or ref number of the AAV from Addgene.
- In Fig EV1-E - please give Tau with SD and statistical comparison. Again in FigEV1H, give mean values with SD.
- It is often not clear in the Fig legends whether the recording were made in perforated conditions or WCR. Is there any difference between the two patch conditions for the fluorescent Glut release response? Some of the perforated recordings were made in 1.3 mM extracellular Ca^{2+} and WCR in 5 mM ext calcium, why? Please, clarify and indicate the recording conditions in each figure legend.
- In Introduction page 4 line 17, I think it is worth mentioning here that a near-linear relationship between Ca^{2+} influx and exocytosis was also observed in vestibular hair cells (Dulon et al., 2009), suggesting a linear synaptic transfer function as a general property of all sensory hair cell ribbon synapses.
- Figure 5 is extremely difficult to read and does not bring essential information. I suggest to move it to supplementary section or reorganize it in a more simple way to only show factors with strong correlations above 0.8 for example.
- Page 23: in the paragraph "heterogeneity of Ca^{2+} channel-release coupling", again the discussion should be extended to other hair cell ribbon synapses such as vestibular hair cells. It should be discussed here that vestibular type I hair cells, known to be connected by high SR low threshold fibers, have specialized their AZ to have a tighter spatial organization of Ca^{2+} channels, with more negative voltage-activation, when compared to auditory IHCs (Vincent et al., 2014).

Referees' comments:

Referee #1:

In this remarkable study by Ozcete and Moser evidence is presented on how hair cells in mammalian ears are able to diversify their synaptic output to better encode sound information. In particular they address a long-standing question on how individual IHC are able to translate a wide range of sound intensities into complementary dynamic ranges in firing rates of SGN. The authors used a state of the art technique, with functional imaging methods to simultaneously track Ca^{2+} influx to IHC (with Rhod FF) and glutamate release (through the genetically encoded glutamate sensor iGluSnFR). They investigated in detail how the voltage dependence of different release sites within a given IHC differed. A correlation is informed, by which Ca^{2+} cooperativity was closer to one on pillar synapses, whereas activation threshold occurred at hyperpolarized potentials on this side of hair cells. Analysis is deep and informative. Conclusions on how these observations could explain high vs low SR fiber segregation are sound. While the technique picked by the authors could not seem to be more appropriate, some methodological questions remain. These questions are laid out in the following paragraphs, with one that stands out refers to the variability in the glutamate sensor signal. But also for the single synapse calcium. How variable are parameters like $V_{1/2}$ or m when imaging the same spot multiple times? Have the authors evaluated this experiment? How variable is $\Delta F/F_0$ on a given glutamate spot? The authors outlined correlations, clearly showing variability for parameters like $V_{1/2}$, m , etc. But functional variability should be contrasted with variability due to detection constraints and/or trial-to-trial variance. Without this information some of the physiological conclusions may be biased or, quite the opposite, loose strength.

We thank the reviewer for the appreciation of our work and the constructive criticism that helped us to improve the MS.

Major points:

-Fig. 1. DF, why not $\Delta F/F_0$? Are those traces in Fig. 1B due to single IHC depolarization? Or average of several?

We utilized ΔF images for better appreciation of the raw data. In response to the reviewer's comment, we also plotted the normalized ΔF images ($\Delta F/F_0$), and they showed similar spatial profiles (new Appendix Fig S2).

The iGluSnFR traces depicted in Fig. 1B are in response to single step depolarizations to most directly inform about the signal to noise. However, each voltage step was repeated twice, and the average traces were used for the further analysis (Fig. 1C-D).

-Methods. For this reviewer it is not intuitive to represent the shape iGluSnFR ROIs. Particularly after applying this maximum entropy thresholding, and seeing images in Fig. 1B. From these representative images it seems as if iGluSnFR fluorescent spots have the size of an entire synaptic bouton (or even larger), whereas glutamate release sites are presumably much smaller. Do fluorescence transients occur over this large areas? How variable are they in pixel number? How compact are ROIs in shape? It would be highly advisable to illustrate the reader with a representative image of individual/average ROIs.

To address reviewer's question about the detection of functional iGluSnFR ROIs, we have now updated Fig. 1, and represented the corresponding ROIs on the ΔF image. Furthermore, we prepared a supplementary figure showing the pipeline for iGluSnFR ROI detection (new Appendix Fig S2). This step-by-step illustration includes the ΔF image (also the $\Delta F/F_0$ image for comparison), median-filtered

image, mask created after the maximum entropy thresholding and the following watershed segmentation. The detected ROIs were variable in size ranging from ~250 pixels to ~1000 pixels. As to the question, on whether the fluorescence changes occur across the entire postsynaptic boutons or are limited to the membrane opposite to the active zone: as evident in Fig. 1A iGluSnFR expression is rather homogenous in the membrane of the spiral ganglion neurons. As the release glutamate spreads from the active zone it activates iGluSnFR across the membrane of the bouton, which we decided to use for boosting the signal to noise ratio. While it would be interesting to study spatial differences of the iGluSnFR responses at different sites of the postsynaptic bouton, this would require future studies with better detection efficiency than obtained with the spinning disk confocal and camera used in the present study.

-Is focus set in Figures 2/3 similarly to Fig. 1?

The focus in Figs. 2 and 3 was set similarly to Fig. 1. In all the cases, the imaging plane was first determined based on the baseline fluorescence of iGluSnFR and test pulses of 20-ms-long step depolarizations were applied to check for the functional response in a given plane.

-Saturation of iGluSnFR is naturally a problem but I do not think that a correlation between the fluorescent signal and C_m could reveal it. If the fluorescent protein could undergo saturation it would most likely occur early during an IHC depolarization, and comparing fluorescence and capacitance in long pulses (>50 ms) would not be necessarily informative. It should be more helpful to establish a correlation between these two variables with short IHC pulses in which the membrane potential was varied (e.g. between -47 and -22 mV).

To address the reviewer's comment, we have performed additional experiments and analysis. As recommended, we applied brief (10-ms) step depolarizations from the holding potential of -87 mV to -57 mV to 23 mV in 10 mV steps (applied in pseudo-randomized order, new Appendix Fig. S5) and simultaneously recorded changes in whole-cell C_m and iGluSnFR fluorescence. We performed the recordings in organs of Corti of P15-19 WT mice injected with AAV9.*hSyn.iGluSnFR* at P6 (perforated patch-clamp, 1.3 mM $[Ca^{2+}]_e$, $n = 11$ boutons, $N = 4$ IHCs from 4 mice). Both the peak and the AUC of iGluSnFR signal positively correlated with the whole-cell ΔC_m in the negative voltage range (from -57 mV to -17 mV; Pearson's $r = 0.57$, $p < 0.0001$, Pearson's $r = 0.59$, $p < 0.0001$, respectively) and in the whole recorded voltage range (from -57 mV to 23 mV; Pearson's $r = 0.58$, $p < 0.0001$, Pearson's $r = 0.61$, $p < 0.0001$, respectively). These findings corroborate our previous observation (Fig. EV1F-G) that iGluSnFR saturation is not a major concern under these conditions.

-One important issue is variability/noise in the fluorescence signal. What's the variability in $V_{1/2}$ of glutamate release (Fig. 3) when imaging multiple times the same spot? Same for the Ca^{2+} dependence of release. How variable is m when repeating the experiment over a single glutamate spot? I believe that a calibration of this variability is required to be certain about conclusions. From looking at individual curves on Figure 3 (although difficult to clearly see), it seems that neither the glutamate nor the calcium signals show monotonic trends but rather an erratic increase and decrease as membrane voltage changes. Could the authors refer to this?

To assess the variability in $V_{1/2}$ of glutamate release, we compared the first $V_{1/2}$ estimates to the second $V_{1/2}$ estimates in the same synapses whereby the two sets of stimuli were separated in time by several minutes over the course of ten minutes. Each $V_{1/2}$ was calculated from the Boltzmann fits using iGluSnFR-AUC measurements for nine depolarization voltages. The first and second $V_{1/2}$ estimates did not show a

significant difference (paired t-test, $p = 0.61$, mean of the first $V_{1/2} = -37.73$ mV, mean of the second $V_{1/2} = -37.34$ mV).

In response to the reviewer's concern on the variability of the peak iGluSnFR signal at a given synapse, we performed additional experiments. We probed the peak iGluSnFR signal by repetitive 20-ms-long step depolarizations from the holding potential of -87 mV to -17 mV, applied in every 20 seconds over 5 mins (new Appendix Fig S4, ruptured patch-clamp, 10 mM intracellular EGTA, 5 mM $[Ca^{2+}]_e$, $n = 5$ boutons, $N = 2$ IHCs from 2 mice). We observed a mild rundown of the ΔF -iGluSnFR over repetitive stimulation: The mean peak amplitude of the first three points showed a 36.75 ± 7.03 % decrease compared to the mean of the last three points (14 step depolarizations over 5 mins), which might also involve rundown of exocytosis.

Supported by our previous work, we assume that in conditions of strong cytosolic Ca^{2+} buffering, the fluorescence change of the low-affinity Ca^{2+} indicator at the AZ faithfully reports synaptic Ca^{2+} influx (Frank *et al.*, 2009; Ohn *et al.*, 2016). Nonetheless, while not likely, we cannot exclude possible effects of Ca^{2+} indicator saturation or Ca^{2+} -induced Ca^{2+} release.

As stated by the reviewer, the individual curves for synaptic Ca^{2+} influx and glutamate release are crowded with all the data points. To ease the reading and allow better judgment of the individual traces, we now plotted every synapse individually including the data points and the fits. Furthermore, we depicted the goodness of fit per fit (r^2) and $V_{1/2}$ estimates (new Appendix Figs. S6 and S7).

-Similarly to the observation on how focus is set, one concerning issue is out of focus images in Figure 4. It is not clear from the narrative how many cases with multiple simultaneous glutamate spots were recorded by the authors, and/or are included in the dataset. But it may be problematic if within the dataset multiple glutamate spots were not imaged at the brightest focal plane. How would these properties differ if glutamate fluorescence was not in focus?

In response to the reviewer's comment, we performed further experiments probing the effect of the imaging plane on the iGluSnFR response. We applied 50-ms-long step depolarizations in 7 different planes separated by $0.5 \mu m$ each (ruptured patch-clamp, 10 mM intracellular EGTA, 5 mM $[Ca^{2+}]_e$, $n = 10$ boutons, $N = 6$ IHCs from 4 mice, new Appendix Fig S3). After finding the plane with the highest signal, we compared it to the signal in planes $\pm 0.5 \mu m$ and $\pm 1 \mu m$ from the defined center plane. We found that the iGluSnFR signal was rather robust towards missing the optimal plane: there were 24.79 ± 4.33 % reduction in the peak iGluSnFR signal for the planes $\pm 0.5 \mu m$ from the center, and 26.83 ± 4.05 % reduction for the planes $\pm 1 \mu m$ from the center.

-How does m change (or not) with respect of the pillar/modiolar axis? Could the authors include a correlation with position as done in Fig. 4B-D?

This is an excellent suggestion: we now added the plot of the position along the pillar-modiolar axis versus the Ca^{2+} cooperativity (m , new Appendix Fig S9). While the Ca^{2+} cooperativity did not show a significant correlation with the position along the pillar-modiolar axis, we found that Ca^{2+} cooperativities ≥ 3 were mainly positioned on the modiolar side.

-Why do the authors think that coupling between whole-cell calcium and glutamate fluorescence is rather linear, whereas this same relation between synaptic calcium and release seems on average supralinear (Fig. 3)? This point requires discussion and/or interpretation. Whole cell calcium could be considered a weighted average of synaptic influx, and so be the calculated ' m '. Shouldn't be this ' m ' equal to the average of single ' m 's calculated from individual synaptic sites?

This an excellent question and we have been thinking much about the reasons, without reaching a conclusion with high confidence. In response to the comment, we have now added discussion that considers the bias of the imaging to exclude the basal cap. As we had explained this was done to safely record from individual synapses in separation. We suspect that there is a fair number of synapses with $m < 2$ in the basal cap, quite densely innervated, which might contribute to the observed discrepancy of the average single m estimates and the m estimated from the whole cell Q_{Ca} .

Minor:

Line 33. ROI detection. Size of Median filtering, please indicate if size used is pixels.

For median filtering, we used `scipy.ndimage.median_filter` (from Scipy package in Python). The size indicates the two-dimensional array of pixels, such as (n,n).

Referee

#2:

EMBOJ-2020-106010, corr. author Prof. Moser
"A sensory cell diversifies its output by varying Ca^{2+} influx-release coupling among active zones"

General summary and opinion about the principal significance of the study, its questions and findings: Auditory inner hair cell (IHC) ribbon synapses have to deal with the great challenge of encoding an extremely wide range of sound intensities, a dynamic range of more than 100 dB. To perform this challenging task, auditory inner hair cells have partitioned their synaptic output sensitivity and dynamic range. Indeed, each IHC form synapses with a pool of 10 to 30 afferent nerve fibers, among which spontaneous activity, acoustic threshold and dynamic range vary widely. The present study tackles one the most important question in sensory neuroscience: what are the functional mechanisms that dictate the synaptic diversity of each type of sensory synapses? To address this question, the authors characterized the synaptic transfer function at each individual IHC synapses by imaging Ca^{2+} -signals and glutamate release during IHC-patch clamp recording. The results showed that variation in the voltage-gating and

spatial-organization of calcium channels at each presynaptic active zone could determine the firing specificity of the auditory nerve fibers. Although the heterogeneity in amplitude and voltage-sensitivity of the calcium responses of the IHC active zones has already been extensively characterized in several previous papers from the same lab, this is the first report characterizing the synaptic transfer function correlating Ca²⁺ influx and glutamate release at each type of post-hearing ribbon synapses. The results nicely confirmed a pillar-modiolar gradient of AZ transfer functions: with pillar AZ, contacting low threshold high-SR fibers, being activated at more negative potentials and having a tighter spatial organization (nanodomain) between Ca²⁺ channels and release sites. This study is technically sound and brings novel and interesting results for the general neuroscience community. The authors convincingly demonstrate how sensory synapses adapt their presynaptic AZ-Ca²⁺ signals to encode a large dynamic range of stimuli. The precise molecular mechanisms controlling this differential voltage gating and Ca²⁺ channel organization at each type of AZs remain now to be determined.

We thank the reviewer for the appreciation of our work and the constructive criticism that helped us to improve the MS.

Specific major concerns essential to be addressed to support the conclusions: Most of my major questions concern the optical detection of glutamate release.

1. The fluorescent Glut responses from synaptic boutons were variable in time and amplitudes as shown in Fig 1B. Could these different responses be also explained by a variability in the kinetics of glut reuptake in each synaptic cleft, i.e. due to a differential expression of EAAT transporter between synapses? I think some comparative measurements should be performed in presence of TBOA to remove possible interference with differential Glut reuptake or buffering.

In response to the reviewer's comment, we performed additional experiments probing the effect of glutamate uptake on iGluSnFR responses. EAAT1 in supporting cells is responsible for glutamate uptake at IHC afferent synapses (Glowatzki *et al.*, 2006). We utilized TFB-TBOA (Tocris, Cat. No. 2532) which selectively inhibits EAAT1 and EAAT2. We perfused 200 μ M of TFB-TBOA while recording iGluSnFR fluorescence in response to 20-ms-long step depolarizations applied every 20 seconds (ruptured patch-clamp, 10 mM intracellular EGTA, 5 mM [Ca²⁺]_e, $n = 5$ boutons, $N = 3$ IHCs from 2 mice, see Fig below). We compared the mean of the three responses before and during the application of TFB-TBOA. We did not find significant differences between the peak or AUC of the iGluSnFR signals before and after TFB-TBOA application (paired t-test, $p = 0.25$ and $p = 0.41$, respectively). Under these experimental conditions, glutamate uptake by EAATs does not seem to have a significant effect on iGluSnFR responses.

2. It is not clear whether all afferent nerve fibers were similarly labeled with the AAV expressing iGluSnFr; in other words, what is the percentage of labeled SGN or synaptic boutons? Also, is the labeling intensity, ie the expression level of the Glut sensor uniform between SGNs? I think this an important issue because the peak-amplitude as well as the integral of the Glut fluorescent responses would greatly be dependent on a differential iGluSnFr density between afferent boutons? This point should be clearly addressed. Ideally, a comparative calibration of the Glut fluorescent responses between pillar and modiolar boutons using extracellular Glutamate-uncaging or Glutamate puffs at various concentrations should be tested.

In response to the reviewer's comment, we have performed further experiments and analysis. We prepared mid-modiolar cochlear cryosections of the injected ear of an P26 mice ~3 weeks after the injection with pAAV.*hSyn*,iGluSnFR virus at P6 and immunolabeled for iGluSnFR (GFP) and IHCs, OHCs and SGNs (parvalbumin). We used parvalbumin as a generic marker for SGNs and quantified iGluSnFR-expressing SGN somata. The percentage of iGluSnFR-expressing SGNs was 99.10 ± 1.46 of the parvalbumin-expressing SGNs in the apical turn, where all the experiments were performed in this study (4 cryosections from an injected cochlea, new Appendix Fig S1). In addition, iGluSnFR expression was also prominent in the SGN somata in the mid- and basal-turns. Taken together, the high transduction efficiency of the virus (>99%) shows almost every SGN innervating IHCs in the apical turn was expressing iGluSnFR.

Furthermore, to exclude the possible contribution of the differential iGluSnFR expression levels on the responses from pillar and modiolar synapses, we compared their baseline iGluSnFR fluorescence and found no significant difference (Mann-Whitney U test, $p = 0.23$, see below Fig.).

3. Because of crowding, the Glut-responses were only measured on a small fraction of the synaptic boutons, those contacting apically the IHCs (as indicated page5). It seems therefore nonrealistic to extrapolate the data in a model that consider only a small fraction of IHC synapses. Is it possible that a classification would have fallen in more than 3 categories if all fibers would have been studied.

This is a fair point and a limitation of our study, which we clearly described, but in response to the reviewer's comment have now discussed this even more explicitly. Nonetheless, in this study, even without the synapses present at the basal cap of the IHC, we found a spatial gradient of synaptic properties such as $V_{1/2}$ and threshold of glutamate release along the pillar-modiolar axis (Fig. 4B-D). Similarly, a previous study (Ohn *et al.*, 2016) on fast Ca^{2+} imaging of IHC synapses found similar spatial differences in the maximal and $V_{1/2}$ of synaptic Ca^{2+} influx along the pillar-modiolar axis both with and without the basal cap synapses. No such spatial gradient was found along the longitudinal axis (Ohn *et al.*, 2016). Taken together, while the analysis of glutamate release from synapses of the basal cap remains an important task for future studies, our data is consistent with a model in which input-output function of IHC synapses forms a gradient along the pillar-modiolar axis. Therefore, while potentially underrepresenting a category, the synapses analyzed in this study likely represent three categories as indicated by the clustering analysis. Whether these presynaptic properties align with the 3 categories of SGNs defined by the classical *in vivo* studies of SGN firing properties (Kiang *et al.*, 1965; Liberman, 1978), and/or by the recent transcriptomic profiles of SGNs (Shrestha *et al.*, 2018; Sun *et al.*, 2018; Petitpre *et al.*, 2018) remains to be established.

4. I wonder whether spontaneous release could be directly recorded by the glut sensor in IHC-contacting boutons, i.e. in absence of depolarizing and voltage-clamping IHCs? In this case, it would have been interesting to directly identify and compare high vs low Spont Release boutons.

This is a great suggestion and we wished the spontaneous rate of release or of SGN firing was readily accessible in these experiments. In response to the reviewer's comment we did additional experiments aiming to identify individual spontaneous release events, but eventually could not convince ourselves that occasionally occurring fluorescence blips potentially representing spontaneous release events provided a sufficient basis for analysis, given that extended times of required imaging would risk bleaching of iGluSnFR.

5. The Pearson's r coefficient of correlation of the synaptic properties, ie for $V_{1/2}$ of Ca^{2+} influx and Glut release is quite low ($r < 0.5$; fig3G). At first glance of Fig 3G, it seems difficult to say that the synaptic factors are correlated. This point should be corrected and discussed. Also in Fig4B-E, the r coefficients are again very low (< 0.5). The conclusions drawn from these figures should be moderated and discussed.

As the cut-offs for the correlation strength can be quite arbitrary in biological systems, we think it is appropriate to state the significant correlations, and further illustrate their relationship with the linear regression model fits along with their confidence intervals. In response to the reviewer's comment, we also calculated the non-parametric ranking-based Spearman's correlation coefficients (see Fig below). Furthermore, we compared the correlation between the $V_{1/2}$ of glutamate release and the $V_{1/2}$ of the fractional activation of the Ca^{2+} channels without the contribution of the single channel current (depicted as $V_{1/2, \text{Pactivation}}$ in Fig. EV6, see Methods for the calculation), and between threshold of glutamate release and $V_{1/2}$ of overall synaptic Ca^{2+} influx. They both showed significant positive correlation (Pearson's $r = 0.46$, $p = 0.0003$; Pearson's $r = 0.54$, $p < 0.0001$).

In addition to the binary comparison of the pillar and modiolar synapses, in which we found significant differences for the given parameters in Fig. 4B-D, we believe that analyzing the synaptic parameters as a function of position along the pillar-modiolar axis provides important information for considering origins of IHC contact-dependent SGN properties. In addition to the gradients supported by the positive or negative correlations along the pillar-modiolar axis, these findings also highlight their heterogeneity, and hint to a more salt-and-pepper like organization rather than a strict bimodal distribution. In response to the reviewer's comment, we added further discussion about this.

6. It looks like that all ruptured WCR recordings were performed with 10 mM intracellular EGTA, thus quenching the Glut release responses of microdomain synapses. To clearly show a differential nanodomain vs microdomain organization for pillar and modiolar synapses, a direct comparison should be made with 1 mM intracellular EGTA in similar recording WCR conditions.

In response to the reviewer's comment, we performed further analysis. To assess the possible quench of the glutamate release by 10 mM intracellular EGTA, we compared the threshold of glutamate release in physiological buffering conditions (perforated patch-clamp, 1.3 mM $[Ca^{2+}]_e$) to the one with high EGTA (ruptured patch-clamp, 10 mM intracellular EGTA, 5 mM $[Ca^{2+}]_e$). The thresholds of glutamate release were comparable (Mann-Whitney U test, $p = 0.42$, see Fig. below), supporting the lack of an obvious glutamate release quench in these high buffering conditions. The reason we performed our whole-cell recordings with high intracellular EGTA was better demarcation of the active zones, as spatially well-

confined “Ca²⁺ hotspots” are required for single-synapse analysis. Our attempts to obtain confined “Ca²⁺ hotspots” with lower Ca²⁺ buffering conditions was confronted with the lower signal-to-noise ratio and/or spread of the Ca²⁺ signal, limiting the detection of single-synapse Ca²⁺ influx. Possibly, a ribbon-binding peptide could be used in parallel to localize the synaptic ribbons and thereby the synapses. However, this was not possible in our dual-color imaging approach due to the overlapping spectrums of the CtBP2-binding peptides, either TAMRA- or Abberior Star 488-conjugated, with Rhod-FF or iGluSnFR fluorescence. While the high EGTA could quench the microdomain synapses, it allowed us to study the transfer functions of individual IHC synapses in separation and, nonetheless revealed their heterogeneity. In response to the comment of the reviewer, we have further discussed this point.

7. In Fig 6, 11 synaptic properties were used for the PCA and K-means clustering. Indicate which ones were selected because I see 15 synaptic properties in Fig 5. Why K was arbitrarily chosen or applied to be 3? I guess because of *in vivo* previously studies reported 3 categories of auditory nerve firing properties. How the data would behave with K = 2 or 4 for example? Could you define what are the three principal components PC1, PC2 and PC3 in Fig 6A-C. Please clarify the analysis of this complex Fig for the general reader.

As indicated in our MS, we applied K-means clustering (K = 3) on the 11th-dimensional space of single-synapse properties (excluding positional and whole-cell information). The excluded parameters include the position along the pillar-modiolar axis, threshold, V_{1/2} and dynamic range of whole-cell Ca²⁺ influx (Q_{Ca}).

As indicated by the reviewer, we chose K of 3 due to the 3 reported categories of SGNs based on the classical *in vivo* studies of SGN firing properties (Kiang *et al.*, 1965; Liberman, 1978), as well as the recent transcriptomic profiles of the SGNs (Shrestha *et al.*, 2018; Sun *et al.*, 2018; Petitpre *et al.*, 2018).

In response to the reviewer’s comment, we performed further analysis by applying K-means clustering with K = 2 and 4 on the 11th-dimensional space of single-synaptic properties (new Appendix Fig S10). We visualized the clusters on the first three principal components and depicted their transfer functions and Ca²⁺ dependencies, as in Fig. 5.

Furthermore, we plotted the correlation map of the first three principal components (PCs) and the synaptic properties (new Appendix Fig S11). The first three PCs showed the highest correlation with the threshold

and dynamic range of synaptic glutamate release, and the threshold of synaptic Ca^{2+} influx. The contribution of the rest of the synaptic properties to the PCs can be seen in Appendix Fig S11.

Minor concerns that should be addressed:

- The title of the study seems too general. Only auditory IHC synapses were studied here; I suggest for example to replace sensory cell by "Auditory inner hair cells diversify their synaptic transfer function by..... "

We wish to keep the title to make the paper attract a broader readership, but we would leave it at the discretion of the reviewers and editor to use the alternative below:

“Hair cells fractionate coding of sound intensity information via afferent synapses with different transfer functions”

- Remove first sentence of the abstract that seems redundant with the next following sentence. Done.
- The introduction and discussion are too long, sometimes repetitive and speculative. These sections should be shortened by a least one page.

We have shortened introduction and discussion.

- Indicate in text legend or methods, the values of the slice thickness of the confocal images. Also clearly indicate in figure 1B the locations of the measured ROI. Done.
- Scale bar 10 μm of Fig 1B is too large - 2 μm is enough. Done.
- Methods page 28: indicate the precise Cat or ref number of the AAV from Addgene. Done.
- In Fig EV1-E - please give Tau with SD and statistical comparison. Again in FigEV1H, give mean values with SD.

Due to the noise in the data, we performed the fits on the average traces, aiming to get an estimate of the synaptic vesicle pool dynamics. This, however, did not allow us to do direct comparison using statistical analysis.

- It is often not clear in the Fig legends whether the recording were made in perforated conditions or WCR. Is there any difference between the two patch conditions for the fluorescent Glut release response? Some of the perforated recordings were made in 1.3 mM extracellular Ca^{2+} and WCR in 5 mM ext calcium, why? Please, clarify and indicate the recording conditions in each figure legend.

We performed the ruptured patch-clamp recordings often with simultaneous Ca^{2+} imaging. As we explained in response to the question 6 of this reviewer, we used 5 mM extracellular Ca^{2+} and high intracellular EGTA concentration to obtain strong, but spatially confined “ Ca^{2+} hotspots”. We have now updated the figure legends.

- In Introduction page 4 line 17, I think it is worth mentioning here that a near-linear relationship between Ca^{2+} influx and exocytosis was also observed in vestibular hair cells (Dulon et al., 2009), suggesting a linear synaptic transfer function as a general property of all sensory hair cell ribbon synapses. Done.

- Figure 5 is extremely difficult to read and does not bring essential information. I suggest to move it to supplementary section or reorganize it in a more simple way to only show factors with strong correlations above 0.8 for example.

In response to the reviewer's comment, we now moved the Fig 5 to the expanded view as Fig EV6.

- Page 23: in the paragraph "heterogeneity of Ca^{2+} channel-release coupling", again the discussion should be extended to other hair cell ribbon synapses such as vestibular hair cells. It should be discussed here that vestibular type I hair cells, known to be connected by high SR low threshold fibers, have specialized their AZ to have a tighter spatial organization of Ca^{2+} channels, with more negative voltage-activation, when compared to auditory IHCs (Vincent et al., 2014). Done.

Dear Tobias,

Thank you for submitting your revised manuscript to The EMBO Journal. Your revision has now been reviewed by the two referees and their comments are provided below. As you can see from their comments, both referees appreciate the introduced changes and support publication here.

I am therefore very pleased to let you know that we will accept your manuscript. Before sending you the formal acceptance letter there are just a few things that should be resolved in a final revision.

- We require a data availability section. This would be the place to enter accession numbers etc. As far as I can see no data is generated that needs to be deposited in a database. If this is correct please state: This study includes no data deposited in external repositories. Please place it after the Materials and Methods and before Acknowledgements
- We don't encourage data not shown (page 6). Maybe either show the data or re-phrase
- I don't think funding information has been added for Özge özcete in the online submission system. Please double check
- Please correct "Methods" to "Materials and Methods". It would also be good to provide a callout to the reagents table somewhere in that section
- Are the Fig 5F curves reused in S10M? if so then please state this in figure legends
- I have asked our publisher to do their pre-publication checks on the paper. They will send me the file within the next few days. Please wait to upload the revised version until you have received their comments.

That should be all. Congratulations on an interesting study!

Best Karin

Karin Dumstrei, PhD
Senior Editor
The EMBO Journal

Further information is available in our Guide For Authors:

The revision must be submitted online within 90 days; please click on the link below to submit the revision online before 25th Jan 2021.

Referee #1:

On ROI configuration, Figure 1 and Appendix Figure 2: The authors made it clearer on how large glutamate ROIs are, and how they were constructed. A range between 250 and 1000 pixels is indicated in the rebuttal letter. From the pixel size of the camera (informed in methods) one could calculate the area of these ROI, in microns. But it may be a good idea to provide this information (are of ROI in microns) in the text.

Appendix Figure S5. Why so little variation in C_m compared to $\Delta F/F$? Why C_m values align like in columns? I presume that those represent values obtained with the same V_m pulse, but still it is not obvious to this reviewer. Please, explain.

It becomes clear that the authors made a very significant effort to address my concerns on the variability of some of the extracted parameters such as $V_{1/2}$. Appendix Figures S4 and S6 also contribute a lot to a better understanding of the experiments.

Besides stability, Figure S4 also provides a good estimate of the glutamate signal fluctuation over time. As I stressed on my previous revision, my concern is mainly variability in the signal when evaluated repeatedly with a constant stimulating pulse. As done for Figure S4.

But it seems less clear for $V_{1/2}$, specially in the box plot included in the rebuttal letter (why not including this data on the main text?). The authors made it clear that the average $V_{1/2}$ does not change over time (at least at two time points). But a good part of this work is about heterogeneity,

and what becomes clear from Figure S4 is that individual measurements of $V_{1/2}$ can vary a lot from one measurement to another. Again, it is clear that differences in $V_{1/2}$ cancel out when repeated over multiple cells, but as each data point in (for instance) Figs. 3 and 4 are single repetitions I wonder if (part of) the pillar-modiolar heterogeneity would change if measurements were repeated in each cell. I believe that this is the point that the authors should address. For instance how comparable are SD for $V_{1/2}$ of Figure 3 and S4. Same concern about m estimation.

Referee #2:

The authors have done an excellent job revising the manuscript. They have adequately responded to most of my concerns. However, I still have a last comment which was raised in my major point 2. An essential control experiment is missing in the manuscript. The authors should demonstrate that iGluSnFR when expressed in cochlear afferent nerve boutons is able to gradually respond to external glutamate application (ie within a normal physiological dose-response curve as previously shown in other studies using brain slices). A similar dose-responses curves (KD) between pillar and modiolar postsynaptic afferent boutons should be demonstrated. These titration-fluorescent experiments should be easily performed by applying, through a small tip glass-electrode placed near the synaptic boutons, brief puffs of glutamate at different concentrations.

Response to editor's comments:

- We require a data availability section. This would be the place to enter accession numbers etc. As far as I can see no data is generated that needs to be deposited in a database. If this is correct please state: This study includes no data deposited in external repositories. Please place it after the Materials and Methods and before Acknowledgements.

Done.

- We don't encourage data not shown (page 6). Maybe either show the data or re-phrase.

Done, added as **new Fig. S5**

- I don't think funding information has been added for Özge özcete in the online submission system. Please double check

All the funding sources were given to Tobias Moser.

- Please correct "Methods" to "Materials and Methods". It would also be good to provide a callout to the reagents table somewhere in that section

Done.

- Are the Fig 5F curves reused in S10M? if so then please state this in figure legends

The curves in Fig 5F are not reused in S10M (now S11). Fig 5F represents the mean responses of the 3 clusters ($K = 3$) and Fig S10M represents the mean responses of the 4 clusters ($K = 4$) of the same dataset.

Referee #1:

-On ROI configuration, Figure 1 and Appendix Figure 2: The authors made it clearer on how large glutamate ROIs are, and how they were constructed. A range between 250 and 1000 pixels is indicated in the rebuttal letter. From the pixel size of the camera (informed in methods) one could calculate the area of these ROI, in microns. But it may be a good idea to provide this information (area of ROI in microns) in the text.

We thank the reviewer for the comment. As stated in the Materials and Methods, the pixel size is 103 nm.

-Appendix Figure S5. Why so little variation in C_m compared to $\Delta F/F$? Why C_m values align like in columns? I presume that those represent values obtained with the same V_m pulse, but still it is not obvious to this reviewer. Please, explain.

As indicated by the reviewer, in Appendix Figure S5 (now S6), the columnar look of the data points is because they are obtained in response to the same step depolarization steps.

-It becomes clear that the authors made a very significant effort to address my concerns on the variability of some of the extracted parameters such as $V_{1/2}$. Appendix Figures S4 and S6 also contribute a lot to a better understanding of the experiments.

We thank the reviewer for the appreciation.

-Besides stability, Figure S4 also provides a good estimate of the glutamate signal fluctuation over time. As I stressed on my previous revision, my concern is mainly variability in the signal when evaluated repeatedly with a constant stimulating pulse. As done for Figure S4. But it seems less clear for $V_{1/2}$, specially in the box plot included in the rebuttal letter (why not including this data on the main text?). The authors made it clear that the average $V_{1/2}$ does not change over time (at least at two time points). But a good part of this work is about heterogeneity, and what becomes clear from Figure S4 is that individual measurements of $V_{1/2}$ can vary a lot from one measurement to another. Again, it is clear that differences in $V_{1/2}$ cancel out when repeated over multiple cells, but as each data point in (for instance) Figs. 3 and 4 are single repetitions I wonder if (part of) the pillar-modiolar heterogeneity would change if measurements were repeated in each cell. I believe that this is the point that the authors should address. For instance how comparable are SD for $V_{1/2}$ of Figure 3 and S4. Same concern about m estimation.

We thank the reviewer for the appreciation and the comment. The data of heterogeneity comes from multiple cells, which should alleviate the concern. The Appendix Figure S4 probed the peak amplitude of iGluSnFR signal in response to the repetitive step depolarizations. It does not provide information on the voltage dependence of release. Therefore, it cannot be compared to the SD of $V_{1/2}$ estimates in Figure 3.

Referee #2:

- The authors have done an excellent job revising the manuscript. They have adequately responded to most of my concerns. However, I still have a last comment which was raised in my major point 2. An essential control experiment is missing in the manuscript. The authors should demonstrate that iGluSnFR when expressed in cochlear afferent nerve boutons is able to gradually respond to external glutamate application (ie within a normal physiological dose-response curve as previously shown in other studies using brain slices). A similar dose-responses curves (KD) between pillar and modiolar postsynaptic afferent boutons should be demonstrated. These titration-fluorescent experiments should be easily performed by applying, through a small tip glass-electrode placed near the synaptic boutons, brief puffs of glutamate at different concentrations.

We thank the reviewer for the appreciation of our work and for their suggestion. As we have shown in the first rebuttal letter, the baseline iGluSnFR fluorescence of the pillar and modiolar synapses were similar in our recorded synapses. Furthermore, in our recording conditions, we did not see a significant effect of EAAT blocker TFB-TBOA on iGluSnFR responses. Therefore, we do not expect to see any differences in the dose-response curves of pillar and modiolar synapses.

Dear Tobias,

Thank you for submitting your revised manuscript to The EMBO Journal. I have now had a chance to take a careful look at everything and all looks good!

I am therefore very pleased to accept the manuscript for publication here. Congratulations on a nice study

With best wishes

Karin

Karin Dumstrei, PhD
Senior Editor
The EMBO Journal

Please note that it is EMBO Journal policy for the transcript of the editorial process (containing referee reports and your response letter) to be published as an online supplement to each paper. If you do NOT want this, you will need to inform the Editorial Office via email immediately. More information is available here: https://emboj.embopress.org/about#Transparent_Process

Your manuscript will be processed for publication in the journal by EMBO Press. Manuscripts in the PDF and electronic editions of The EMBO Journal will be copy edited, and you will be provided with page proofs prior to publication. Please note that supplementary information is not included in the proofs.

Should you be planning a Press Release on your article, please get in contact with embojournal@wiley.com as early as possible, in order to coordinate publication and release dates.

If you have any questions, please do not hesitate to call or email the Editorial Office. Thank you for your contribution to The EMBO Journal.

Corresponding Author Name: Tobias Moser

Manuscript Number: EMBOJ-2020-106010R